
# Airborne and ground-based observations of ammonium nitrate dominated aerosols in a shallow boundary layer during intense winter pollution episodes in northern Utah

Alessandro Franchin[1,2], Dorothy L. Fibiger[1,2,Δ], Lexie Goldberger[3,*], Erin E. McDuffie[1,2], Alexander Moravek[4], Caroline C. Womack[1,2], Erik T. Crosman[5], Kenneth S. Docherty[6], William P. Dube[1,2], Sebastian W. Hoch[5], Ben H. Lee[3], Russell Long[7], Jennifer G. Murphy[4], Joel A. Thornton[3], Steven S. Brown[1], Munkhbayar Baasandorj[5], and Ann M. Middlebrook[1]

[1]NOAA Earth System Research Laboratory (ESRL) Chemical Sciences Division, 325 Broadway, Boulder, CO 80305, USA
[2] Cooperative Institute for Research in Environmental Sciences (CIRES), University of Colorado, Boulder, CO 80309, USA
[3] University of Washington, Department of Atmospheric Sciences, Seattle, WA 98195, USA
[4] Department of Chemistry, University of Toronto, Toronto, ON, M5S 3H6, Canada
[5] University of Utah, Department of Atmospheric Sciences, Salt Lake City, UT 84112, USA
[6] Jacobs Technology, Inc., RTP, NC 27709, USA
[7] Environmental Protection Agency (EPA), Durham, NC 27709, USA
,Δ Now at California Air Resources Board, Monitoring and Laboratory Division, Sacramento, CA 95811, USA.
* Now at Pacific Northwest National Laboratory, Richland, WA 99354, USA.

*Correspondence to*: Alessandro Franchin (alessandro.franchin@noaa.gov)

**Abstract.** Airborne and ground-based measurements of aerosol concentrations, chemical composition and gas phase precursors were obtained in three valleys in northern Utah (U.S.A.). The measurements were part of the Utah Winter Fine Particulate Study (UWFPS) that took place in January-February, 2017. Total aerosol mass concentrations of $PM_1$ were measured from a Twin Otter aircraft, with an Aerosol Mass Spectrometer (AMS). $PM_1$ concentrations ranged from less than 2 $\mu g\ m^{-3}$ during clean periods to over 100 $\mu g\ m^{-3}$ during the most polluted episodes, consistent with $PM_{2.5}$ total mass concentrations measured concurrently at ground sites. Across the entire region, increases in total aerosol mass above ~ 2 $\mu g\ m^{-3}$ were associated with increases in the ammonium nitrate mass fraction, clearly indicating that the highest aerosol mass loadings in the region were predominantly attributable to an increase in ammonium nitrate. The chemical composition was regionally homogenous for total aerosol mass concentrations above 17.5 $\mu g\ m^{-3}$, with 74±5% (average ± standard deviation) ammonium nitrate, 18±3% organic material, 6±3% ammonium sulfate, and 2±2% ammonium chloride. Vertical profiles of aerosol mass and volume in the region showed variable concentrations with height in the polluted boundary layer. Higher average mass concentrations were observed within the first few hundred meters above ground level in all three valleys during pollution episodes. Gas phase measurements of nitric acid ($HNO_3$) and ammonia ($NH_3$) during the pollution episodes revealed that in Cache and Utah Valley, partitioning of inorganic semi-volatiles to the aerosol phase was usually limited by the amount of gas phase nitric acid, with $NH_3$ being in excess. The inorganic species were compared with the ISORROPIA thermodynamic model. Total inorganic aerosol mass concentrations were calculated for various decreases of total nitrate and total ammonium.



For pollution episodes, our simulations of a 50% decrease in total nitrate lead to a 46±3% decrease in total PM$_1$ mass. A simulated 50% decrease in total ammonium lead to a 36±17% µg m$^{-3}$ in total PM$_1$ mass, over the entire area of the study. Despite some differences among different locations, our results also showed a higher sensitivity to decreasing nitric acid concentrations and the importance of ammonia at the lowest total nitrate conditions. In the Salt Lake Valley, both HNO$_3$ and NH$_3$ concentrations controlled aerosol formation.

## 1 Introduction

Intense wintertime air pollution from particulate matter affects numerous locations in the United States (Chen et al., 2012; *Lurmann et al.*, 2006) and around the world (Bessagnet et al.,2005; Gwaze et al., 2007; Ricciardelli et al., 2017; Wang et al., 2014). In the U.S., pollution from fine particles (PM$_{2.5}$) has been decreasing in the past decades. However, PM$_{2.5}$ concentrations in Salt Lake City and surrounding valleys remain among the highest nationwide (EPA, 2017). High PM$_{2.5}$ episodes in these areas occur predominantly during winter, when persistent cold-air pools (PCAPs) form as a result of high-pressure ridges (Whiteman et al. 2014, Lareau et al. 2013). These events increase in frequency and severity during snow covered periods (Green et al., 2015). Under PCAP conditions, emissions of pollutants and precursor gases are trapped in a shallow boundary layer, as vertical mixing is limited. Precursor gases react chemically, increasing the production of PM$_{2.5}$ via secondary pathways. Almost every year, wintertime concentrations of PM$_{2.5}$ exceed the limit of 35 µg m$^{-3}$ in a 24 h average in the Logan, Ogden-Clearfield, Salt Lake City and Provo-Orem Core-Based Statistical Areas (CBSAs, EPA Air Quality Statistics by City, 2016), leading to violations of the National Ambient Air Quality Standards (NAAQS) set by the Environmental Protection Agency (EPA). This air quality issue affects almost 80% of the population of the state of Utah, and has been associated with increased risk for stroke, heart disease, lung cancer, and both chronic and acute respiratory diseases [*Beard et al.*, 2012; *Pope et al.*, 2006, WHO, 2016].

Chemical composition measurements of PM and their precursor gases, together with a detailed understanding of the meteorology are key to identifying sources and developing mitigation strategies. Previous studies showed that ammonium nitrate (NH$_4$NO$_3$) dominates the aerosol mass during pollution episodes (Hansen et al., 2010; Kelly et al., 2013; Kuprov et al., 2014; Long et al., 2003; Long et al., 2005a; Long et al., 2005b; Mangelson et al.,1997; Silva et al., 2007). Analysis of ground-based data suggests and that its formation, at least in the Salt Lake Valley and Cache Valley is likely to be nitric acid limited (Kuprov et al., 2014; Mangelson et al., 1997). The Great Salt Lake is a potential source of chloride in the area, and chloride may also play a role during pollution episodes, although its role is much less clear. Hansen and co-workers found chloride (in the form of sodium chloride) to be a minor part of the PM$_{2.5}$ mass fraction (1.3% on average over the period of their study January 22–31, 2007 in Lindon, UT; Hansen et al., 2010). However, Kelly and co-workers, using two different factor analysis techniques, found that, after ammonium nitrate, ammonium chloride could also be a significant source of secondary wintertime PM$_{2.5}$ (Kelly et al., 2013).



Formation of PM$_{2.5}$ results in part from a complex interaction of chemical production mechanisms with boundary layer meteorology (Baasandorj et al., 2017a). Previous studies carried out in wintertime in San Joaquin Valley (CA) highlighted the critical role of nocturnal chemical production of nitrate aloft in the residual layer (i.e. that region decupled from the stable nocturnal boundary layer influenced by previous day surface emissions). Those studies suggested that nighttime nitrate formation within the residual layer was a major contributor to surface-level PM$_{2.5}$ concentrations (Watson et al., 2001; Brown et al., 2006; Chow et al., 2006; Lurmann et al., 2006, Prabhakar et al., 2017). Studies in other regions of the U.S. have reached similar conclusions regarding the role of nighttime processing aloft as a source of winter soluble nitrate (Stanier et al., 2012; Kim et al., 2014). Similarly, Baasandorj and co-workers found evidence in their ground-based observations in the Salt Lake City area for nitrate formation occurring at night and early morning photochemistry in the upper levels of a PCAP (Baasandorj et al., 2017a). Although the residual layer is inferred to be a critical region for understanding process level atmospheric chemistry leading to winter ammonium nitrate in polluted regions, few air quality studies have directly probed residual layer chemical composition using an instrumented aircraft in a polluted winter boundary layer (Brown et al., 2013, McDuffie et al., 2018).

In this work, we present results of the Utah Winter Fine Particulate Study (UWFPS), an intensive measurement campaign combining airborne and surface-based measurements that took place in Salt Lake City and surrounding valleys in January and February 2017 (Figure 1 and Figure S1). In particular, we focus on the chemical and physical aerosol properties measured by an aerosol mass spectrometer (AMS) and an ultra-high sensitivity aerosol spectrometer (UHSAS) aboard the National Oceanic and Atmospheric Administration (NOAA) Twin Otter research aircraft. The flights took place from January 16[th] to February 12[th] and were based out of the Salt Lake City International Airport, with flight plans extending up to Cache Valley, across the Great Salt Lake, and over the Salt Lake and Utah Valleys. Flight plans included missed approaches at several local airports in the three main study regions to probe the vertical structure of the concentrations of aerosols and precursor gases. The NOAA Twin Otter aircraft was equipped with a suite of instruments that allowed comprehensive measurements of the aerosols and the gas phase precursors that contribute to the pollution episodes in the area.

The goal of this analysis is to achieve a better understanding of the processes that drive the conversion of precursor vapors into aerosol particles. We compared measured inorganic gas and particle composition to a thermodynamic model and used the model output to probe sensitivities to changes in that composition that are useful to developing effective PM$_{2.5}$ control strategies.



## 2 Methods

The airborne measurements, carried out using the NOAA Twin Otter aircraft, were designed to characterize pollution episodes with a particular focus on ammonium nitrate aerosol formation. The aerosol measurements from the aircraft were compared with ground-site data of $PM_{2.5}$, $PM_{10}$, and non-refractory $PM_1$ (NR-$PM_1$) obtained by the Utah Department of Air Quality (UDAQ) and the U. S. Environmental Protection Agency (EPA). The thermodynamic model ISORROPIA was used to compare predicted aerosol to vapor partitioning with the measurements.

### 2.1 Twin Otter Aircraft Instrumentation

### 2.2.1 Aerosol Measurements

Aerosols were sampled using a perpendicular near-isokinetic inlet described and characterized previously for aircraft measurements (Perring et al., 2013). The inner diameter of the secondary diffuser of the inlet was modified from the original inlet (Ø=2.5 mm in the modified version) to accommodate a 3 liter per minute (l min$^{-1}$) total inlet flow and the lower cruise speed of the Twin Otter (~250 km h$^{-1}$). The total inlet flow was chosen in order to limit sampling line losses due to diffusion and evaporation. The sampled air downstream of the inlet probe was split into a 37 cm sampling line to the UHSAS and a 153 cm sampling line to the AMS with an excess flow of about 2.8 l min$^{-1}$ that was discarded. The excess flow was monitored with a linear flow element and a calibrated pressure transducer (Honeywell 480-5439-ND).

The Aerosol Mass Spectrometer (AMS, Aerodyne Research Inc., Billerica, MA) measures the chemical composition of the non-refractory aerosol particles in the 70 to 850 nm range (NR-$PM_1$) (Canagaratna et al., 2007; Drewnick et al., 2005; Jayne et al., 2000). The working principle of the AMS is to 1) focus ambient aerosols in a vacuum chamber with an aerodynamic lens, 2) evaporate the aerosol particles by impacting them onto a 600 ºC surface, and 3) ionize the evaporated molecules with electron-impact ionization and analyze the resulting fragments with a mass spectrometer to obtain information on the chemical composition of aerosols. For the majority of sampling, the AMS collected bulk mass spectra (MS mode). Additionally, aerosol size distributions are obtained by operating the AMS in particle time of flight (pToF) mode wherein mass spectrometer signals are monitored as a function of the time required for particles of different sizes to traverse the length of the vacuum chamber. The NOAA AMS deployed on the Twin Otter aircraft (TO-AMS) was equipped with a pressure-controlled inlet to maintain a constant mass flow rate into the instrument as the aircraft changed altitude (Bahreini et al., 2008). The TO-AMS was also equipped with a light scattering module (LS) that provided a more accurate size information in the range 200-1000 nm and on the collection efficiency (Liao et al., 2017). A CE of 1 was used for this analysis based on the light scattering (see SI).

The TO-AMS was operated in MS mode (6 seconds, with 4 s open to sample and 2 s closed for background correction) and pTOF mode (4 seconds) in 27 sub-cycles of 10 seconds total. We measured in LS mode for 30 seconds every 4.5 minutes. The aircraft AMS measurements are all reported in units of micrograms per standard cubic meter of air, where standard conditions





are 273 K and 1 atm. Details of the AMS calibration procedures for the UWFPS are presented in the SI. The 10 s average limit of detection for nitrate ($NO_3^-$), ammonium ($NH_4^+$), organic species (Org), sulfate ($SO_4^{2-}$), chloride ($Cl^-$) and for the total aerosol mass were 0.04, 0.09, 0.33, 0.03, 0.07, and 0.38 µg m$^{-3}$ respectively. The uncertainty in the total AMS mass concentrations is estimated to be 20% (Bahreini et al., 2009). It is important to note that refractory species such as sea and / or lake salt (mostly

sodium chloride), road salt (mostly magnesium chloride), dust (mostly alkali salts and silicon oxides), and black carbon (from diesel exhaust or wood combustion) are not routinely measured with the AMS.

The Ultra-High Sensitivity Aerosol Spectrometer (UHSAS, Droplet Measurement Technologies, Longmont, CO) was used to measure the dry size distributions of aerosol particles in the range of 70 to 1000 nm. The working principle of the UHSAS is

based on optical-scattering. The aerosol particles enter the instrument and scatter the light produced by a solid-state laser (1054 nm, 1 kW/cm$^2$). Mirrors capable of collecting light over a large solid angle (22°–158°) direct the scattered light to two different photodiodes: an avalanche photodiode for detecting the smallest particles and a low-gain PIN photodiode for detecting particles in the upper size range. The signals generated by the scattered light are used for particle sizing since the amount of light scattered correlates strongly with particle size (Cai et al., 2008). UHSAS data during the UWFPS were recorded every 3 sec

and are reported here per standard cubic centimeter (273 K and 1 atm). The relative humidity in the sampling line was calculated to be less than 30%. The UHSAS inlet flow was monitored with a linear flow element and a calibrated pressure transducer Honeywell 480-5439-ND) in order to correct for variations in the inlet flow (Brock et al., 2011). The uncertainty during UWFPS was 2% on the sizing, 14% on the number concentration, 20% on the total surface and 35% on the total volume.

**2.2.2 Gas Phase Measurements**

On the Twin Otter aircraft, we obtained measurements of several gas phase species, including those relevant for inorganic gas-aerosol partitioning in the region: nitric acid ($HNO_3$), hydrochloric acid (HCl), and ammonia ($NH_3$). We used these aircraft gas phase species, as well as data from aerosol phase nitrate, chloride, and ammonium to calculate the ratios of gas over the total ammonium or total nitrate for estimating which species is the limiting reagent in ammonium nitrate formation and for the input to thermodynamic modeling with ISORROPIA (see section 2.3).

The University of Washington high-resolution time-of-flight iodide adduct ionization mass spectrometer, hereafter referred to as HRToF-CIMS, was used to measure a suite of reactive inorganic gaseous halogen (HCl, $Cl_2$, $Br_2$, $ClNO_2$, etc.) and nitrogen oxide ($HNO_3$, $N_2O_5$, HONO, etc.) species. The major components of the CIMS were described previously (Lee et al., 2014) with upgrades implemented prior to UWFPS to improve in-flight calibration and sampling protocol (Lee et al., 2017). Ambient

air was drawn in to the CIMS at a rate of ~22 liters per minute through a 39 cm long 1.6 cm inner diamter (ID) polytetrafluoroethylene (PTFE) inlet, resulting in a mean residence time of about ~0.21 seconds. The CIMS instrument subsamples 2 slpm from the centerline of the main flow traveling through the 1.6 cm ID inlet with the remainder of the flow being pulled radially through a circular slot around and downstream of the ion molecule reaction (IMR) region entrance, to



minimize sampling air that has been in contact with the inlet surface. Instrument background levels were determined every 60 seconds for a duration of 5 seconds by overflowing the IMR with ultra high purity nitrogen (UHP $N_2$). The UHP $N_2$ was introduced at the entrance of the IMR region, which represents the dominant source (>80%) of the sum of the background levels originating from the inlet and IMR region under field conditions. The molecular formulae of the compounds listed above

are readily identified given its mass resolving power (4500-5500 ppm (Junninen et al. 2010), with no known spectrally interfering species. For nitric acid ($HNO_3$) and hydrogen chloride (HCl), the calibration uncertainty for both is about 30%, while their limits of detection (LOD), determined by accounting for their calibration coefficients and variabilities in their background levels were 60 and 160 ppt, respectively.

A Continuous-Wave Quantum Cascade Tunable Infrared Laser Differential Absorption Spectrometer (QC-TLDAS, Aerodyne Research Inc., Billerica, MA; Tevlin et al 2017) was used to measured ammonia ($NH_3$) concentrations from the Twin Otter aircraft. Prior to installation, the instrument was optimized for a lower total weight allowing its operation on the Twin Otter aircraft. Furthermore, it was equipped with a pressure control system to account for changing ambient pressure with altitude. Ambient air was aspirated at a flow rate of 4 l min$^{-1}$ through the multi-pass absorption cell (0.5 l, 76 m effective path length)

and $NH_3$ absorption was detected at 965.3 cm$^{-1}$. In order to minimize adsorption of ammonia on the inlet walls, an additional bypass flow rate of 16 l min$^{-1}$ was introduced to purge the 50 cm long inlet tubing. In addition, the inlet was heated to approximately 40°C to further reduce tubing wall effects. Evaporation of ammonia from $NH_4NO_3$ aerosols was minimized by removing the particulate matter using a PFA (Teflon) virtual impactor upstream of the QC-TILDAS absorption cell. The uncertainty of the QC-TLDAS during UWFPS was 150 ppt (1σ). The limit of detection (3σ) was 450 ppt at the 1 Hz sample

frequency and 90 ppt for a 30 s averaging interval.

Nitrogen oxides (NO, $NO_2$) total reactive nitrogen ($NO_y$) and ozone ($O_3$) were measured at 1 second time resolution using the NOAA custom-built, nitrogen oxide cavity ring down spectrometer (NOxCaRD). The instrument directly measures $NO_2$ by optical absorption at 405 nm, and quantitatively converts NO and $O_3$ into $NO_2$ by reaction with excess $O_3$ or NO, respectively,

in two separate channels (Fuchs et al., 2009; Washenfelder et al.,2011). A fourth channel converts $NO_y$ to NO and $NO_2$ by thermal dissociation in a quartz inlet at 650 °C and subsequently converts NO to $NO_2$ in excess $O_3$ (Wild et al., 2014). The measurement precision is 50 pptv or better, but zero uncertainty can be as large as 0.2 ppbv.

### 2.2.3 Meteorological parameters

A commercial met probe (Avantech) measured meteorological parameters (ambient temperature, pressure, relative humidity with respect to liquid water, wind speed and wind direction), the global positioning satellite (GPS) location including altitude above sea level, and aircraft parameters (heading, pitch, and roll). Wind data were compromised for some flights, making only





partial coverage available. The aircraft GPS altitude above sea level was converted into altitude above ground level using USGS data (https://gis.utah.gov/data/elevation-and-terrain/).

## 2.2 PM$_{2.5}$ ground-based measurements

As part of the Utah air monitoring network, UDAQ continually monitors PM$_{2.5}$, PM$_{10}$, nitrogen oxides or NO$_x$ (NO and NO$_2$),

ozone (O$_3$) and meteorological parameters according to EPA guidelines and operates 24-h Federal Reference Method (FRM) samplers for PM$_{2.5}$ for NAAQS compliance. Here, we used PM$_{2.5}$ data from two UDAQ sites: Hawthorne, the main air monitoring station for Salt Lake City (40.734397N, 111.8721W), situated on the Salt Lake Valley floor at ~ 1306 m above sea level (ASL) and North Provo (40.253826N, 111.663311W), situated in a central location of the Utah Valley at ~ 1402 m ASL. We also used PM$_{2.5}$ and NR-PM$_1$ data obtained by the EPA at a temporary sampling station located in Logan (41.758875N,

111.815122W), on the Utah State University campus in the Cache Valley at ~ 1405 m ASL. These three sites are located in three sub-basins in proximity of the Great Salt Lake, in northern Utah.

## 2.3 Thermodynamic calculations using ISORROPIA

Cold temperatures and the presence of both nitric acid and ammonia are conducive to forming ammonium nitrate in the aerosol phase. To examine phase partitioning of these compounds for the flight data, we used the thermodynamic model ISORROPIA

(v2.1) (Fountoukis and Nenes, 2007; Nenes et al., 1998). We run ISORROPIA in "forward" mode, in which the total (gas + particulate) measured concentrations of various inorganic species are input into the model along with meteorological data (temperature and relative humidity). From these input parameters, concentrations for each phase are calculated using the van't Hoff equation assuming thermodynamic equilibrium. The Kelvin effect is neglected in the model calculations and all the aerosol particles are assumed to have the same composition.

The inorganic species used for input to ISORROPIA were simplified here to the available flight data for total nitrate (HNO$_3$+NO$_3^-$), total ammonium (NH$_3$+NH$_4^+$), total chloride (HCl+Cl$^-$), and sulfate (SO$_4^{2-}$). The composition of the particle phase (NO$_3^-$, NH$_4^+$ and SO$_4^{2-}$) was from the TO-AMS, therefore NR-PM$_1$. As the AMS does not quantitatively measure refractory sodium or its salts (e.g., sodium chloride, nitrate, or sulfate) under normal operating conditions, the model input for

particulate sodium was set to zero with the assumption that this does not affect the partitioning of the acidic gases when their corresponding ammonium salts. In other words, the equilibrium vapor pressures of the inorganic acids examined here are constrained only by their ammonium salts and the vapor pressures of the acidic gases over their sodium salts are considered to be negligible. Because nitric acid is known to displace chloride from sodium, potassium, calcium, and magnesium salts (Bondy et al., 2017; Metzger et al.; 2006; Sudheer and Rengarajan, 2015; Trebs et al., 2005), aerosols containing these refractory

species could act as a sink for HNO$_3$ and source of HCl in this environment. It also important to note that the term "total nitrate" in this manuscript refers to gas phase plus PM$_1$ nitrate, but may exclude a non-negligible part of nitrate from the coarse mode.



The ISORROPIA output was compared directly with UWFPS measurements. We then performed several simulations designed to test the model response to decreasing the total nitrate and/or ammonium in order to simulate how much a reduction in total nitrate and/or total ammonium would affect the total aerosol mass. When we compared the total NR-PM$_1$ mass at different

conditions, we used only the inorganic mass, as we did not have enough information on to the organic constituents to include them in the model.

## 3 Measurement results

### 3.1 Comparison of ground sites and airborne observations

The UWFPS field intensive measurement period encompassed two complete major pollution episodes (Episode #1 and #2)

and the beginning of a third minor pollution episode (Episode #3, Figure 2, left). All of the three ground sites: Logan, in Cache Valley; Hawthorne, in Salt Lake Valley and North Provo, in Utah Valley showed similar increases in PM$_{2.5}$ mass concentrations as a function of time. However, we observed some slight differences that are probably related to the availability of precursor gases and oxidants at the three sites as well as the various basin topographical and meteorological features (Baasandorj et al., 2017a).

The 2016/17 winter had several multi-day pollution episodes and the PM$_{2.5}$ temporal evolution of each individual episode during the UWFPS study period was similar to those from past years (Figure S2). PM$_{2.5}$ levels in the Salt Lake Valley (SLV, in red) often reach a plateau after an initial increase regionally, whereas PM$_{2.5}$ concentrations continue to increase in Utah and Cache Valleys as long as the PCAP persists. Often a PCAP erodes first in Utah Valley and persists the longest in Cache Valley (, which is surrounded by mountains and sheltered more effectively from mixing induced by large-scale winds. Similar changes

in aerosol mass and volume as a function of time were also observed with the aircraft instruments over the study region. TO-AMS NR-PM$_1$ measurements near the ground sites (within 8 horizontal km and altitudes less 50 m above ground level, AGL) during the study period correlated well (R$^2$=0.82) with the PM$_{2.5}$ ground site data measured at the same time and agreed within the measurement uncertainties (Figure 2, right). Hence, aerosol measurements from the aircraft were representative of and consistent with the ground observations. During the first pollution episode, both the aircraft and ground-based aerosol mass

concentrations were the highest in Cache Valley (~70 μg m$^{-3}$) and the lowest in Utah Valley (~10 μg m$^{-3}$). During the second pollution episode, the highest aerosol mass concentrations were observed in Utah Valley (~70 μg m$^{-3}$) and varied for Salt Lake and Cache Valley over the course of the episode (40 – 90 μg m$^{-3}$). During UWFPS, PM$_{2.5}$ was on average around 70% of PM$_{10}$ in the three valleys (Figure S3).

### 3.2 Aerosol chemical composition during polluted and clean conditions

For the purposes of this analysis, we define clean conditions, when the total aerosol mass is ≤ 2 μg m$^{-3}$ and polluted conditions when the total aerosol mass is > 17.5 μg m$^{-3}$. Our definition of polluted conditions follows the classification used by Whitemann



et al. (2014) that links a total aerosol mass > 17.5 μg m$^{-3}$ to PCAP conditions. NR-PM$_1$ composition differed substantially between polluted and clean conditions (Figure 3). During polluted conditions (44% of the Twin Otter data) the dominant species was ammonium nitrate (74±5%, mean ± standard deviation), while organic, sulfate (mostly ammonium sulfate), and chloride (in the form of ammonium chloride) constituted a minor fraction of the mass (18±3%, 6±3%, and 2±2%, respectively).

The largest aerosol mass concentrations of ammonium nitrate occurred at temperatures below the ice frost point. During clean conditions (56% of the data), the organic fraction dominated the mass of the aerosols (50±13%), ammonium nitrate was the second most abundant species (31±9%) followed by ammonium sulfate and chloride (13±7% and 6±6% respectively). During pollution episode #2, we observed at ground level an average increase of PM$_{2.5}$ over time of about 11 μg m$^{-3}$ day$^{-1}$. The increase due to aerosol phase ammonium nitrate was 8.2±0.4 μg m$^{-3}$ day$^{-1}$, and the remaining contribution of 2.8±0.1 μg m$^{-3}$ day$^{-1}$ was

due to the other chemical species according to the mass fractions indicated above. Furthermore, the change in chemical composition was uniform across the three different valleys (Figure 4) and across all other areas in the studied region (Figure S4) both for airplane and for ground-based measurements (Figure S5), although detailed ground-based composition measurements were limited to the site in the Cache Valley. We can therefore say that aerosol mass concentrations were predominantly due to growth in ammonium nitrate across the entire region. This uniformity occurs despite heterogeneity in

emissions, with relatively large urban emissions from Salt Lake City and relatively large agricultural emissions in the Cache Valley.

A comparison of the total aerosol mass measured with the AMS with the dry aerosol volume measured with the UHSAS is shown in Figure S6. The scatterplot gives an average slope of 1.8 g cm$^{-3}$, which is comparable to the density of ammonium

nitrate (1.72 g cm$^{-3}$), within the combined measurement uncertainty. The aerosol-phase ion balance (Figure 5) shows that the aerosols were completely neutralized, indicating that the measured nitrate, sulfate and chloride are predominantly in the form of their corresponding ammonium salts (ammonium nitrate, ammonium sulfate, and ammonium chloride). The data points in orange have a low ammonium nitrate mass fraction and are below the 1:1 line (more acidic). Those correspond to measurements in the plume of the U.S. Magnesium plant (Figure S1) where we observed a high organic (74.0±1.2%), high

chloride (6±0.6%) and low ammonium mass fraction (3±0.6%). It is important to note that those AMS measurements of the U.S. magnesium plume reported in the current dataset carry a larger uncertainty than the rest of the dataset. The organic component of the aerosol mass was high (up to 50 μg m$^{-3}$) and it is likely to be underestimated as CE = 1 was used (CE is < 1 for aerosols that are not ammonium nitrate and usually assumed = 0.5 for organic aerosol). In the plume, the reported nitrate mass is over estimated as the peak at m/z = 30, that is normally attributed to the NO fragment, showed a contribution from

organic fragments, probably CH$_2$O or C2H6. The plume presented a high aerosol chloride mass, in the form of ammonium chloride at concentrations up to 3.9 μg m$^{-3}$.



### 3.3 Contribution of residential wood combustion to the aerosol mass and other organic aerosol sources to the total aerosol mass

Figures 3 and 4 show that the mass fraction of organic material in the particulate phase on a regional basis was generally less than 20% during polluted conditions. There are very few data points in these aircraft observations containing higher organic

mass fractions at the highest aerosol mass concentrations, which is consistent with sources of wood combustion organic aerosol varying widely (Shrivastava et al., 2007; Kleindienst et al., 2010; Woody et al., 2016). Hence, this organic aerosol fraction of roughly 20% places an upper limit on the average direct contribution (i.e., from primary aerosol emissions or formation of secondary organic aerosol from wood combustion VOCs) of wood combustion to the regional total aerosol mass.

The organic signals in the mass spectra were examined to further estimate the contribution from wood combustion. Prior studies using an AMS instrument indicate that a potential marker of wood combustion is signal at the mass-to-charge ratio ($m/z$) of 60, which is a major ion fragment present in spectra from the known biomass burning marker species, levoglucosan (Alfarra et al., 2007). The fraction of the organic mass spectrum that is at $m/z$ 60 (termed $f_{60}$) is about 8% in spectra from levoglucosan (Alfarra et al., 2007) and is often significantly larger than the background value of 0.3% (or 0.003) for air masses

that are influenced by biomass burning (Cubison et al., 2011). During wintertime evenings, wood combustion is known to be a major contributor to the measured aerosol mass in several European suburban/urban sites, where $f_{60}$ was measured on average to be 0.028 in Roveredo, Switzerland (Alfarra et al., 2007), 0.02 in Grenoble, France (Favez et al., 2010), and slightly less than 0.02 in Augsberg, Germany (Elasser et al., 2012). Here we measured a median value (with 25[th] and 75[th] percentile) for $f_{60}$ of 0.0035 (0.0031 − 0.0042) for the entire study during all conditions, indicating that the contribution of wood combustion to the

organic mass fraction was generally fairly low compared to the European sites. The lower panel of Figure 6 shows the distribution of $f_{60}$ for this study (light red) compared previous studies where the influence of biomass burning was negligible (green, blue and dark red lines). Most of the spectra from our study had $f_{60}$ close to the background value of 0.003, however the distribution shows a somewhat higher tail for our $f_{60}$ values (17% of the data). By comparing with the average fractions from the European wintertime studies, we estimate that the fraction of organic aerosol from wood combustion in these higher

$f_{60}$ cases was about 30% of the total organic mass, i.e., about 6% of the total NR-PM$_1$.

Low values of $f_{60}$ could be due to the initial wood combustion aerosol emissions having low $f_{60}$ (Heringa et al., 2012; Schneider et al., 2006) or aging in the atmosphere causing $f_{60}$ to decrease (Corbin et al., 2015; Ortega et al., 2013) before the aerosol was measured on the aircraft. Hence, relatively higher signals from aged aerosols could potentially indicate conversion of the fresh,

primary biomass burning aerosols to lower $f_{60}$ due to aging. The common marker for aged, oxidized organic aerosols is the peak at $m/z$ 44, which corresponds to the $CO_2^+$ fragment (Allan et al., 2004). $m/z$ 44 correlates with the elemental oxygen/carbon ratio (O:C) (Aiken et al., 2007, 2008; Canagaratna et al., 2015), and indicates organic aerosol aging (Ng et al., 2010; Cubison et al., 2011). A plot of $f_{44}$ (i.e., the ratio of $m/z$ 44 to the total organic aerosol mass) versus $f_{60}$ for our study is shown along with Cubison and co-workers' previous data in the upper panel of Figure 6. The data of freshly emitted biomass burning aerosols





shown in Cubison et al. (2011) tended to have high $f_{60}$ (0.02 to 0.04) with relatively low $f_{44}$ (about 0.05) that changed to lower $f_{60}$ (<0.02) and higher $f_{44}$ (0.15 to 0.23) as the aerosols aged. For the UWFPS study, however, $f_{60}$ was generally low and $f_{44}$ varied considerably.

Since $f_{44}$ was not consistently high, the organic aerosol was not always highly oxidized, indicating that peaks other than $m/z$ 60 were possibly important in the organic aerosol for the UWFPS dataset. Another marker to consider is $f_{57}$ (ratio of $m/z$ 57 to the total organic signal in the AMS) which corresponds to a characteristic fragment from hydrocarbon-like organic aerosol (HOA) from lubricating oil and/or diesel fuel and related to traffic emissions (Canagaratna et al., 2004; Zhang et al., 2005). We observed $f_{57}$ median values of 0.009, 0.012 and 0.010 for Cache, Salt Lake and Utah Valley, respectively. The reported $f_{57}$

for the New York City diesel bus exhaust in Canagaratna et al. (2004) and for the Pittsburgh HOA in Zhang et al. (2005) was 0.081 and 0.075, respectively. This indicates that the some of the organic fraction of NR-PM$_1$ from the UWFPS study could have been from traffic emissions and was highest in the Salt Lake Valley as expected based on the higher urban density there. Note that this marker peak at $m/z$ 57 could also be present in railroad and aircraft emissions.

Previous ground-based observations in this region indicate significant spatial and temporal variability in the contribution of residential wood combustion (Cropper et al., 2018; Mouteva et al., 2017; Baasandorj et al., 2017b). The data reported here indicate that organic material from wood combustion is not the dominant mass fraction on a regional scale. However, smoke concentrations could contribute to most of the aerosol mass at smaller scales or in near source regions (Kelly et al., 2013). Wood combustion may also have an indirect effect on the formation of inorganic aerosol via secondary pathways involving

oxidation of VOCs emitted from this source. Other tracers of various organic aerosol emissions and secondary formation (e.g., black carbon, carbon monoxide) were not obtained from the aircraft during this project, making it difficult to determine the most important organic aerosol sources for the region.

### 3.4 Vertical profiles

Vertical profiles show that the aerosol volume and mass concentration was variable as function of height. Within 1 km above

ground level (AGL) we observed in several individual vertical profiles, layers of aerosols at different heights in each location, at different times of the day at different evolutionary stages of the pollution episodes (Figure S7). These changes in aerosol concentrations as function of height were measured by both the UHSAS and the AMS (Figure 7). The possible causes for these layers could be related to a difference in sources, losses, or to the many transport and mixing mechanisms that occur in complex terrain as discussed in Hoch et al. (in preparation), that are difficult to constrain with our limited vertical measurements during

the study (Baasandorj et al., 2018).

Figure 8 shows the average vertical profiles of NR-PM$_1$ and ammonium nitrate mass fraction for Cache Valley, Salt Lake Valley and Utah valley during Episode#2. On average, Cache and Salt Lake Valley showed either a constant or a slightly



increase in aerosol mass with altitude, up to about 200 m AGL in Cache and up to about 400 m AGL in Salt Lake Valley. Utah Valley show a general decrease in aerosol mass with altitude, shallower in the first 300 m AGL and steeper between 300 and 450 m AGL. Cache and Salt Lake Valley also show a decrease in aerosol mass with altitude from 200 to 450 m AGL and from 400 to 700 m AGL respectively. The average vertical profiles of all the three valleys show a very clear transition where the

aerosol mass is very low above 500 m AGL in Cache Valley, 700 m AGL in Salt Lake Valley and 650 in Utah Valley. The average vertical profiles of the ammonium mass fraction show for all the three valleys that 75% or more of the aerosol mass is ammonium nitrate and that this is true for the first 300 m AGL in Cache Valley, 500 m AGL in Salt Lake Valley and 300 m AGL in Utah Valley. Above those heights, the variability in the chemical composition of the aerosol particles increases and the ammonium nitrate fraction decreases. The marked difference aloft, where the aerosol mass is very low is also characterized

by a fraction of ammonium nitrate of 50% or less, similar in composition to what we observed at lower altitudes in clean conditions.

Interestingly, in some vertical profiles (Figure S7), especially at intermediate mass loadings (from 2 to 17 µg m$^{-3}$), higher mass loadings correspond to larger ammonium nitrate mass fractions (Figure S7 c, e, n, o). However, in other vertical profiles,

changes in mass loadings do not correspond to a change in chemical composition (Figure S7 d, j, l, m). The detection of larger fractions of ammonium nitrate at higher altitudes (Figure S7 f, n, o) might indicate formation of nitrate aloft. A clear change in chemical composition is always detectable between 500 and 1000 m AGL for all the locations. This change is related to the transition from the boundary layer, affected by regional sources, to the free troposphere affected by long-range transport, and consists of a clear change from ammonium nitrate dominated aerosol to sulfate and organic dominated aerosol, concomitant

with a decrease in aerosol concentration.

### 3.5 Limiting Reagent of ammonium nitrate formation

A straightforward method of determining which gas phase precursor (ammonia or nitric acid) is the limiting reagent in ammonium nitrate aerosol formation for the region is to compare the ratios of each species in the gas phase to the total: e.g., ammonia divided total ammonium $NH_3/(NH_3 + NH_4^+)$ and nitric acid divided total nitrate $HNO_3/(HNO_3 + NO_3^-)$. Figure 9

shows these ratios for Cache, Salt Lake and Utah valleys and Figure S8 for the other areas of the study. For all three valleys, the nitric acid gas phase fraction is nearly always smaller than the gas phase fraction of ammonia. Because a stoichiometric, 1:1, ratio of nitric acid to ammonia is needed to form ammonium nitrate, nitric acid is clearly limiting the formation of ammonium nitrate. The fraction of gas phase nitric acid also tends to decrease sharply with increasing total mass concentrations, indicating that the available nitric acid partitions efficiently to the aerosol phase at relatively moderate aerosol

mass concentrations. In contrast, the fraction of gas phase ammonia is larger, irrespective of particle mass concentration and decreases less steeply with increasing mass loadings (Figure 9).



The relative importance of nitric acid in controlling the total aerosol mass varies between the three valleys. For the Cache Valley, the gas phase fraction of nitric acid is always near zero due to relatively large amounts of total ammonium. The gas phase fraction of nitric acid in the Utah Valley is frequently near zero, indicating that slightly more nitric acid (or less ammonia) is available there compared to the Cache Valley. In contrast, the gas phase fraction of ammonia for the Salt Lake Valley is

more frequently smaller than Cache and Utah Valley, especially at the largest total aerosol mass concentrations. Thus, during the most polluted conditions in the Salt Lake Valley, gas phase ammonia is relatively less abundant, indicating a possible transition from nitric-acid-limited towards ammonia-limited ammonium nitrate formation as pollution episodes evolve.

## 4 Thermodynamic modeling of ammonium nitrate

A comparison between the measurements and the ISORROPIA model was carried out to investigate how effectively the

measured partitioning between the aerosol and gas phases are represented by the thermodynamic model. We found that gas phase ammonia, gas phase hydrochloric acid, aerosol ammonium, and aerosol nitrate agree well with the model predictions ($r^2$ = 0.984, 0.962, 0.984, 0.997 respectively ) while gas phase nitric acid and aerosol chloride (as ammonium chloride) are not well represented by the model ($r^2$ = 0.555 and 0.161, respectively; Figure S9, Table S1)**.** It is important to note, however, that these two species have relatively lower mass concentrations and therefore have larger uncertainties associated with them. For

example, the median $HNO_3$ concentration for polluted events was 0.6 µg m$^3$, comparable to 3 times the detection limit of the HRToF-CIMS. Furthermore, the detection limit is typically determined in low particulate nitrate conditions. Therefore, in "polluted conditions" often less than 2% of particulate nitrate evaporating in the inlet system could explain the amount of gas-phase $HNO_3$ detected by the HRToF-CIMS. The modeled gas phase nitric acid was neither consistently higher nor lower than measured, with no clear dependency on temperature nor relative humidity. The aerosol chloride concentrations predicted by

ISORROPIA were generally larger than measured. This overestimate of particle chloride by ISORROPIA has been also observed with other datasets (Haskins et al., 2018) and may affect the scenarios described below, especially for the effect of reducing total nitrate concentrations.

### 4.1 Simulated response to decreasing precursors

We carried out simulations of different scenarios using ISORROPIA (Figures 10, S10 S11 and S12). Figure 10 shows the

simulated decrease in total aerosol mass for two cases: one in which the total nitrate was reduced by a factor of two while keeping the total ammonium constant (blue), and one in which the total ammonium was reduced by a factor of two while keeping the total nitrate constant (orange). Reduction in total nitrate led to approximately a proportional decrease in total aerosol concentrations during polluted conditions and across all locations. Conversely, a decrease in ammonium gave a more variable response. In some regions, a 50% decrease in total ammonium gave a decrease in aerosol mass of 50% or close to it,

while in other regions there was no change. Cache Valley did not respond to a decrease in total ammonium. In fact, most of the orange points for the simulated mass lay on the 1:1 line, indicating no decrease in total aerosol mass compared to the mass





at current conditions. On the contrary, there was a proportional decrease in aerosol mass when the total nitrate was decreased (blue points along to 1:2 line). These results confirm that in this valley, there was an excess of gas phase ammonia and the $NH_4NO_3$ formation chemistry is nitrate limited. The Salt Lake Valley responded to a reduction of either total nitrate or total ammonium (both blue and orange points are closer to the 1:2 line than the 1:1 line), suggesting that limiting either species

could reduce aerosol formation. Despite the mass balance analysis showing excess reduced nitrogen, the thermodynamic model shows the total aerosol mass to be nearly proportionally sensitive to reductions in the either reagents. The Utah Valley, similarly to the Salt Lake Valley, showed a response for the reduction of total nitrate or total ammonium, except for mass loadings lower than 20 µg m$^{-3}$ where a cluster of data points shows no response to the reduction of total ammonium. The area above the Great Salt Lake and Cache Valley were the least sensitive to the reduction of total ammonium while all other locations show varying

responses for aerosol nitrate and chloride to the decrease in total ammonium. Interestingly, the aerosol chloride in the Salt Lake Valley and over the Great Salt Lake showed a slight increase for the decrease in total nitrate, pointing to a change in aerosol composition from ammonium nitrate to ammonium chloride. It is important to note that ISORROPIA generally over-predicts aerosol chloride (see Figure S9) and decreases in total nitrate would potentially decrease the available total chloride from acid displacement reactions on the alkaline salts. Hence, the simulated total reduction of nitrate carries a larger

uncertainty.

A comprehensive view of the sensitivity of total aerosol mass to reductions in total ammonium and nitrate is shown in Figure 11a. This sensitivity study is based on 100 ISORROPIA simulations where data for the entire region (the whole dataset) were used as the input; the total nitrate and/or the total ammonium were decreased at steps of 10% and for each step the average

total aerosol mass was calculated from the model output. For example, the 20% contour line (orange) shows the interpolation of the model output that leads to a 20% reduction in total mass. This plot suggests that while reductions in both total ammonium and total nitrate would be effective overall, the most effective strategy to reduce the total aerosol mass in the region is through reduction of the total nitrate. The orange 10% contour line is reached with total nitrate reductions of about 15% without total ammonium reductions (bottom axis) whereas the total ammonium reduction would have to be more than 20% to reach this

contour line without total nitrate reductions (left axis). However, both reagents must decrease in order to achieve a reduction of total aerosol mass larger than 40% relative to observed conditions. The importance of ammonium, once the total aerosol mass has been decreased, is shown as one-dimensional plots in Figure 11b and 11c where it is shown the simulated decrease in total aerosol mass by decreasing only total nitrate or only total ammonium.

Note that these predictions are somewhat simplified since the model includes total chloride (HCl+Cl$^-$) where the aerosol chloride is ammonium chloride and changes in the amount of either total ammonium or total nitrate can cause additional effects for ammonium chloride partitioning. Furthermore, the model tends to over-predict the amount of aerosol ammonium chloride compared to observations. In the real environment, decreases in total nitrate would likely cause decreases in available total





chloride because less displacement of chloride by nitric acid should occur from refractory salts such as sodium, calcium, or magnesium chlorides.

**5 Conclusions**

During the Utah Winter Fine Particulate Study (UWFPS), we measured gas precursors and aerosols from a NOAA Twin Otter aircraft over Salt Lake City and the surrounding valleys from January 16th to February 12th, 2017. The $PM_{2.5}$ mass loadings for the region varied from below the AMS detection limit of 0.38 to 72.4 µg m$^{-3}$ (2nd and 98th percentiles). The most severe pollution episodes occurred during persistent cold air pool (PCAP) periods, when emissions were trapped near the surface in the three valleys studied (Salt Lake, Cache, and Utah). Aircraft measurements provide spatial and temporal evolution of the non-refractory aerosol chemical composition, as well as the aerosol spatial and temporal evolution.

The particle chemical composition during polluted conditions (aerosol mass > 17.5 µg m$^{-3}$) was markedly different compared to clean conditions (aerosol mass ≤ 2 µg m$^{-3}$). The aerosols during polluted conditions were characterized by large fractions of ammonium nitrate (74±5%) while the fractions of organic, sulfate (mostly ammonium sulfate) and chloride (in the form of ammonium chloride) were smaller (18±3%%, 6±3% and 2±2% respectively). This composition was uniform across all three valleys. The relatively low fraction of organic material indicated that wood combustion played a minor role in the direct formation of secondary aerosols during polluted conditions, as the amounts of the traditional biomass burning marker for the AMS spectra ($f_{60}$) were similar to the ones measured in locations where biomass burning is negligible. However, we cannot completely exclude the role of wood combustion in the regional aerosol mass concentrations as aging in the atmosphere can cause the AMS markers to appear similar to background levels and measurements of other biomass burning tracers such as acetonitrile and black carbon were not available to constrain the AMS observations. During clean conditions, the organic fraction dominated the mass of the aerosols (50±13%), ammonium nitrate still remained the second most abundant species (31±9%) followed by sulfate and chloride (13±7% and 6±6% respectively).

Vertical profiles of the polluted region often showed varying layers of aerosol concentrations alternating. These variations did not always correspond to marked changes in chemical composition, although sometimes we observed a larger ammonium nitrate aerosol mass fraction below the top of the cold air pool. Whether this observation is due to meteorology or chemistry remains unclear. Above the cold air pool, the aerosol mass concentrations dropped considerably and the composition was characteristic of free tropospheric aerosols from long-range transport with relatively more sulfate and organic material and less ammonium nitrate.

Using gas phase measurements of nitric acid and ammonia, we calculated the fraction of each species in the gas phase relative to the total (gas + aerosol). For all three valleys, the fraction of nitric acid in the gas phase was smaller than the gas phase



fraction of ammonia, indicating that the formation of ammonium nitrate in the aerosol phase was limited by the amount of nitric acid in the gas phase. The relative importance of nitric acid in controlling aerosol mass was largest for the Cache Valley, intermediate for the Utah Valley, and less important for the Salt Lake Valley. The gas phase fraction of ammonia in the Salt Lake Valley is often less than one at the largest total aerosol mass concentrations, indicating that gas phase nitric acid

potentially becomes less limiting as pollution episodes evolve under PCAP conditions.

The largest aerosol mass concentrations of ammonium nitrate occurred at temperatures below the ice frost point. The comparison between measurements and the output of ISORROPIA showed agreement for ammonia, hydrochloric acid, aerosol ammonium, and aerosol nitrate. Measured nitric acid and the observed total chloride partitioning is not well represented by the

model, with aerosol ammonium chloride consistently larger in the model. However, for these species the uncertainties in the model are larger than the other chemical species because their measured ambient concentrations were much smaller.

When we simulated a 50% decrease in total (gas + aerosol) ammonium using ISORROPIA we found that Cache Valley did not show a decrease in total aerosol mass compared to measured total ammonium levels, while Salt Lake and Utah valleys

showed a decrease of 44% and 37% respectively. When a 50% decrease in total nitrate was simulated, a 44% decrease in total aerosol mass was observed in all the three valleys, indicating that in all the locations $NH_4NO_3$ formation is controlled by nitric acid availability. However, with the evolution of the pollution episode under PCAP conditions over time, ammonium nitrate formation in the system appeared to become less nitric acid limited for Salt Lake Valley and Utah Valley, while Cache Valley always had excess ammonia.

**Acknowledgments**

The authors would like to acknowledge Utah Division of Air Quality for their support and collaboration during the study. We thank the NOAA Aircraft Operations Center for their dedication and professionalism, particularly the pilots Robert Mitchel and Jason Clark for skillful navigation in complex terrain and challenging meteorology. We thank Frank Erdesz for helping with the installation of the UHSAS, Kyle Zarzana for helping with the calibration and installation of the NOxCARD, Patrick

Veres for helping with the calibration of the HRToF-CIMS. We thank Tim Morphy and Teledyne for the loan of the $PM_{2.5}$ monitor at the Logan site. We thank Jon Abbatt at the University of Toronto and with the Canadian Aerosol Research Network, funded by the Canada Foundation for Innovation, for lending the UHSAS.

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

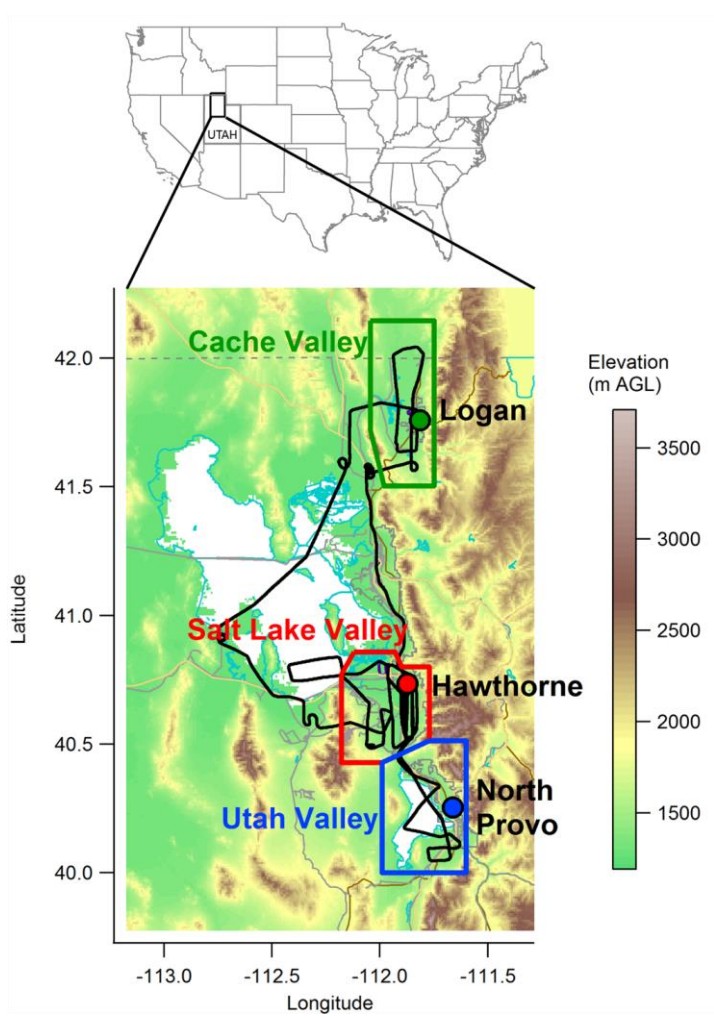

**Figure 1: Map showing the area studied during the UWFPS. Top: map of the United States with the region of interest of this study highlighted by a black rectangle. Bottom: detail of the area of interest colored by elevation as shown in the color bar. In black is shown a typical flight track. The Cache Valley study region is outlined in green, Salt Lake Valley in red and Utah Valley in blue.**
10 **The circles with the same color pattern and labeled in black represent the locations of the ground sites with data presented here.**





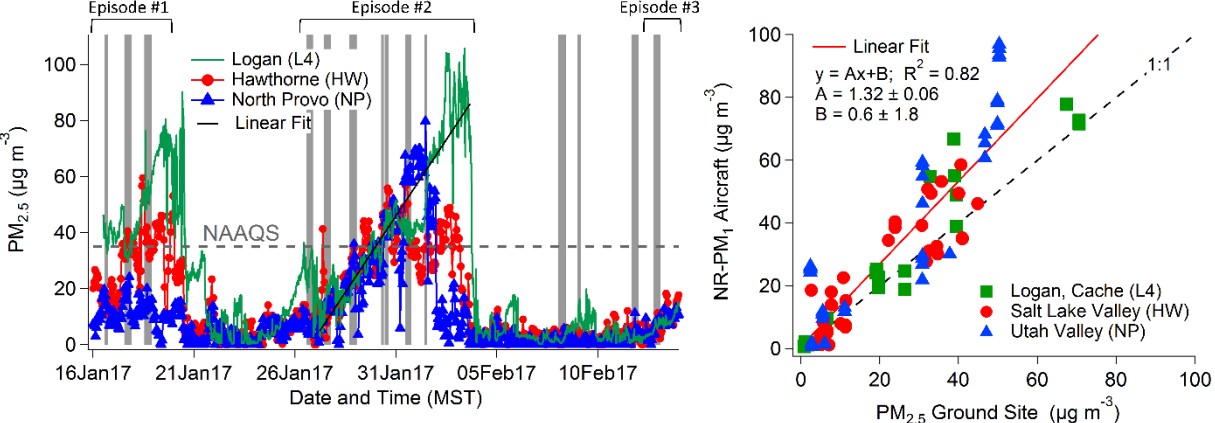

**Figure 2: PM$_{2.5}$ measurements from three ground sites during UWFPS. Left: time series of PM$_{2.5}$ measured in Logan (L4), Cache Valley in green; at Hawthorne Elementary School (HW), Salt Lake Valley, in red; and in North Provo (NP), Utah Valley, in blue. The grey shaded areas correspond to the times of Twin Otter flights; the horizontal dashed line indicates the National Ambient Air Quality Standard (NAAQS) of 35 ug m$^{-3}$ for 24-hr averaged PM$_{2.5}$. The black solid line is a fit over all the data of increase in PM$_{2.5}$ during Episode#2. The slope is 10.89 ± 0.08 µg m$^{-3}$ day$^{-1}$ (slope ± 95% CI). Right: comparison of the non-refractory (NR) PM$_1$ measured with the AMS from the Twin Otter for altitudes lower than 50 m AGL during the missed approaches at the airports closest to the ground site measurements. The dashed line is the 1:1 line and the red solid line is a linear fit; correlation coefficient, fit coefficients and their 95% CI are reported in the legend.**



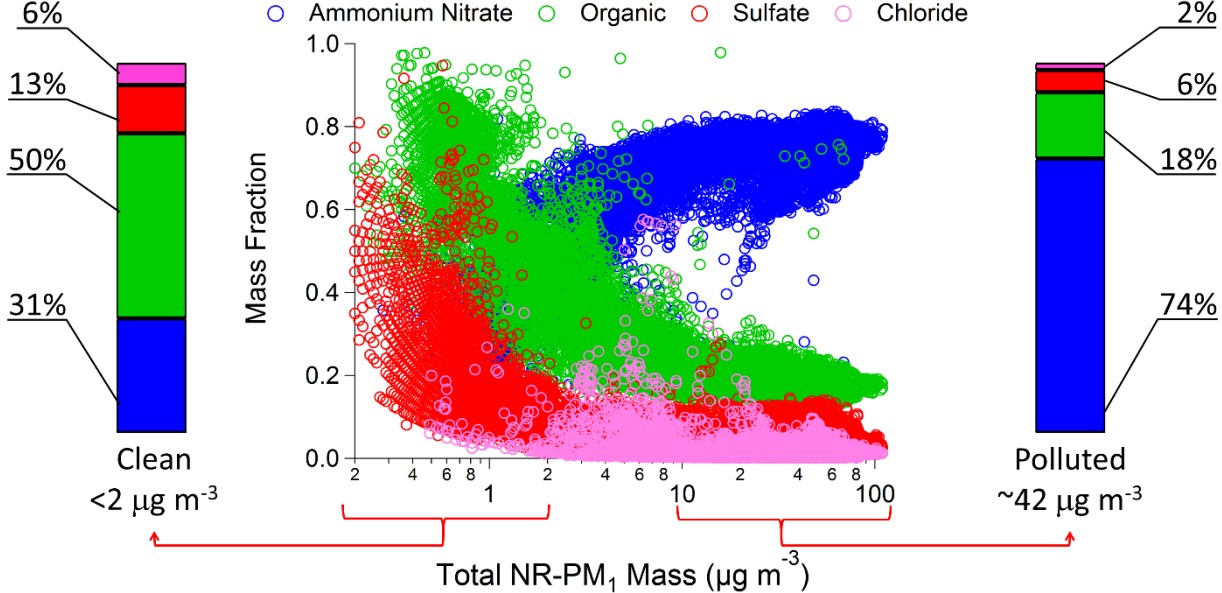

**Figure 3: Center: Aerosol mass fraction as a function of the total mass of NR-PM₁ measured from the Twin Otter.**
**Ammonium nitrate is in blue, organic in green, sulfate and chloride in red and pink, respectively. The bar chart on the**
**left corresponds to the average of the mass fractions when the total aerosol mass is <2 µg sm⁻³ (clean conditions). The**
**bar chart to the right corresponds to the average of the mass fractions when the total aerosol mass is >17.5 µg sm⁻³**
**(polluted conditions) and altitude is <900 m AGL (below the boundary layer).**





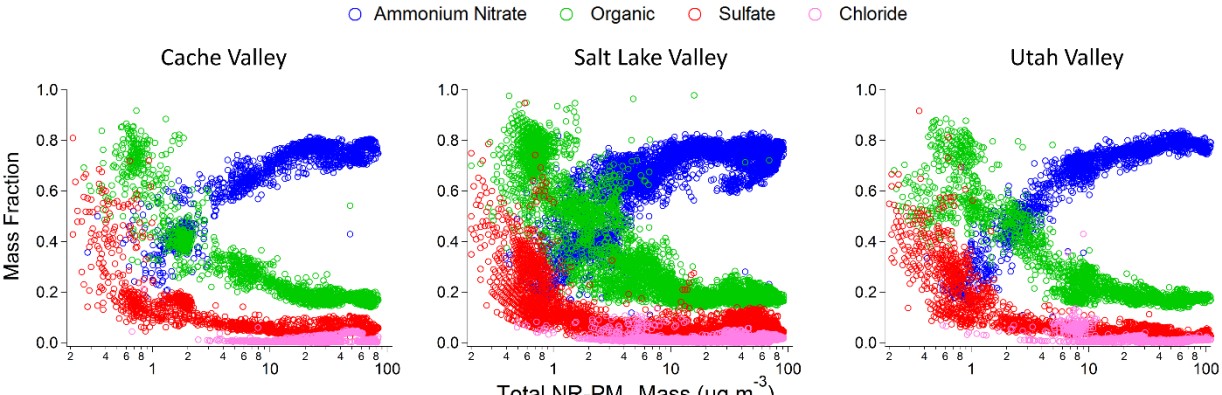

**Figure 4: Aerosol mass fraction as a function of total NR-PM$_1$ mass for each of the three main study areas measured from the Twin Otter. Color-code same as Figure 3.**

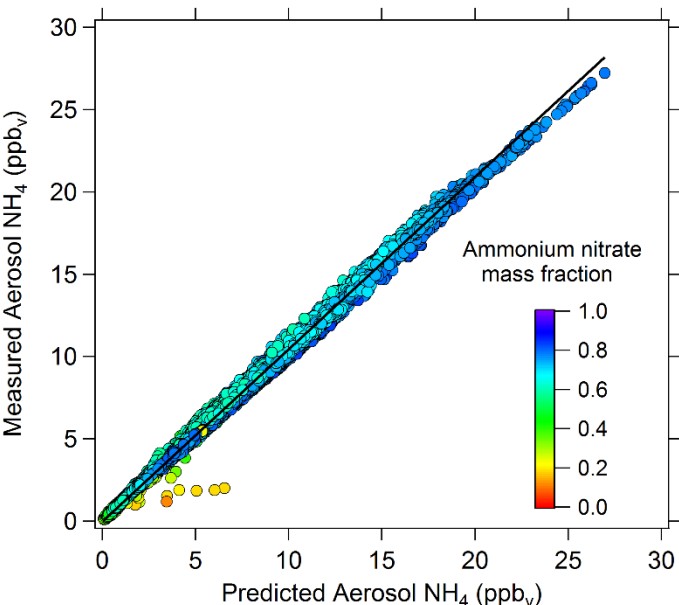

**Figure 5: Aerosol ion balance from the AMS measurements. On the x-axis is the calculated mass of ammonium for fully neutralized aerosols ($NH_4^+ = 18 \times (NO_3^-/62 + SO_4^-/96 \times 2 + Cl^-/35.45)$). On the y-axis is the measured ammonium ($NH_4^+$) in the aerosol particles. The black line is the 1:1 line, where points on the line correspond to neutralized aerosols and below the line are acidic aerosols. Data are color-coded by the ammonium nitrate mass fraction.**




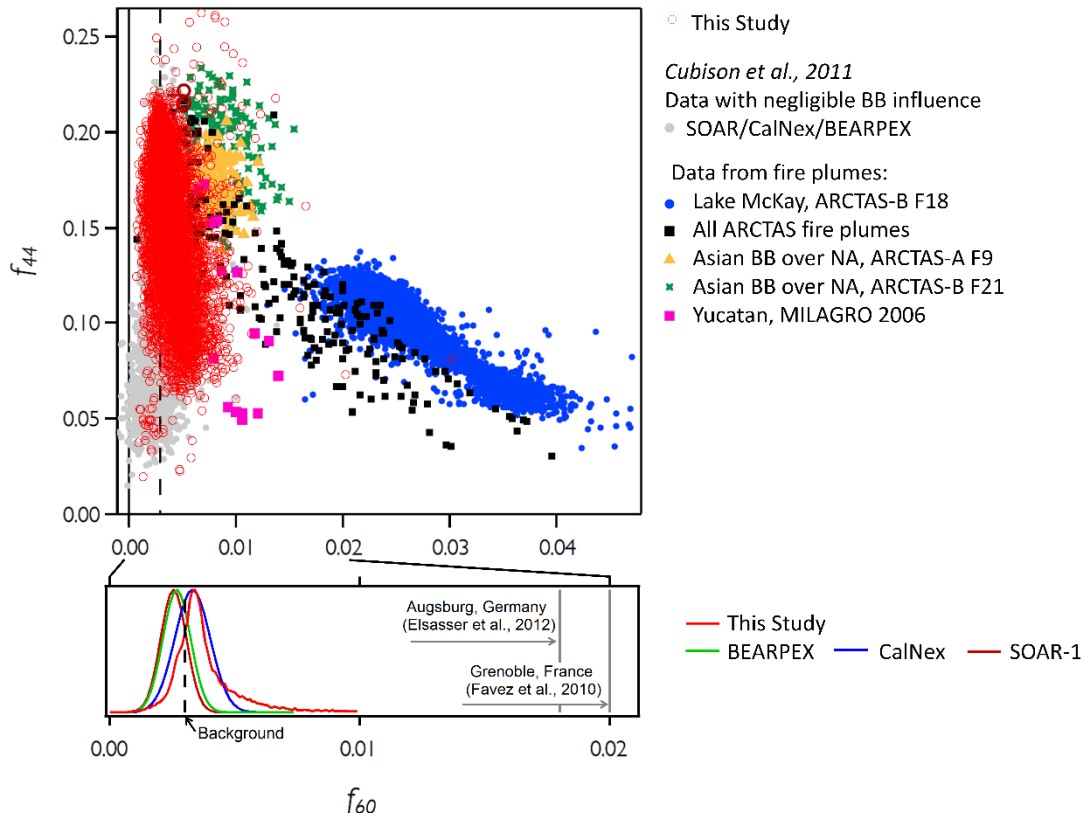

**Figure 6: Top: fraction of the organic fragment at** *m/z* **= 44 divided by the total organic mass (*f₄₄*) as a function of the fraction of the organic fragment at** *m/z* **= 60 divided by the total organic mass (*f₆₀*). *f₄₄* and *f₆₀* are commonly used as markers for oxidized organic aerosol and for freshly emitted biomass burning aerosol, respectively The red open circles are the data measured during the entire UWFPS. The other symbols show data from previous studies (see Cubison el al., 2011 and references therein), with colored (non-red) points for the biomass burning (BB) plumes and grey points for data with negligible BB influence. The dashed black line corresponds to 0.003, the AMS background value for *f₆₀*. Bottom: frequency distribution of *f₆₀* values for this study (light red) and for previous studies with negligible BB influence (green, blue and dark red lines). The vertical grey solid lines show *f₆₀* values for locations where the influence of wood combustion on organic aerosol was strong.**





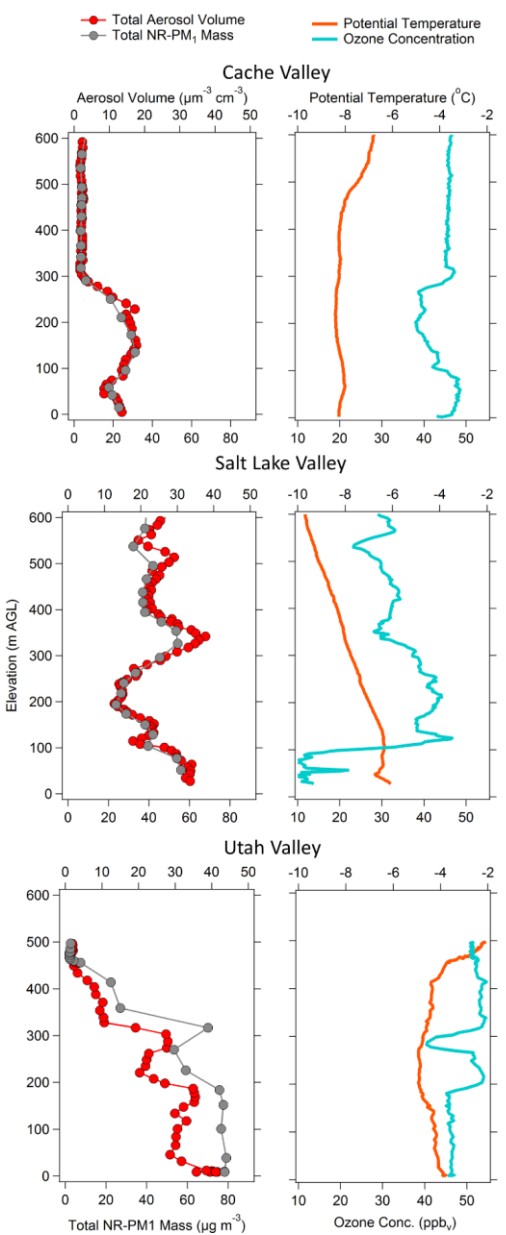

**Figure 7: Comparison between selected vertical profiles. On the left is shown the aerosol volume, measured with the UHSAS (in red) and NR-PM₁ mass measured with the AMS (in grey) as function of elevation in m AGL. On the right: are shown vertical profiles of ozone concentration (light blue) and potential temperature (orange) for the same three missed approaches. The missed approach in Cache Valley took place on Jan 28 at 18:38 MST at the Logan-Cache Airport (LGU); the one in Salt Lake Valley on Jan 28 at 20:22 MST at Salt Lake City International (SLC), and the one in Utah Valley on Jan 30 at 13:24 MST at the Provo Municipal Airport (PVU).**





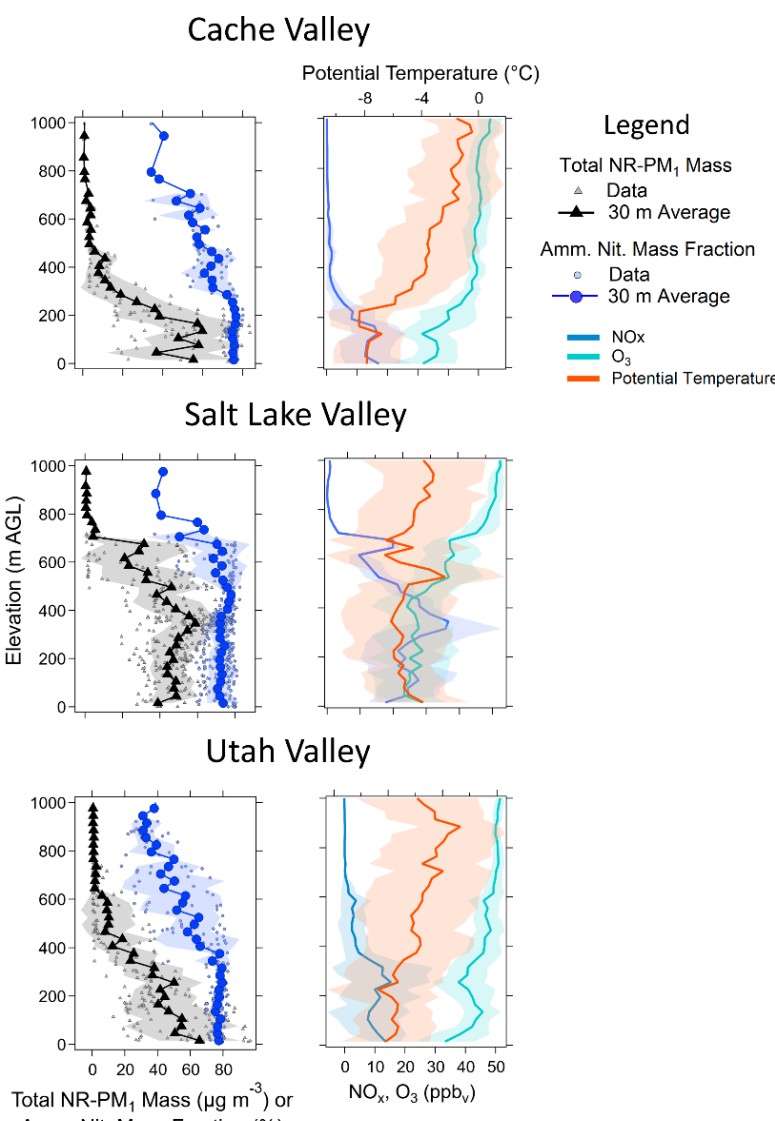

**Figure 8: Left: average vertical profiles of NR-PM1 mass (black triangles) and ammonium nitrate mass fraction (blue circles) measured during Episode#2. The triangular small grey markers and the small light blue circles are data used in the averages for all the missed approaches at the same airports as in Figure 7. The black triangles and dark blue circles are averages over 30 meters with their associated standard deviations. Right: corresponding vertical profiles of NOx (blue lines), O3 (aqua lines) and potential temperature (orange lines).**





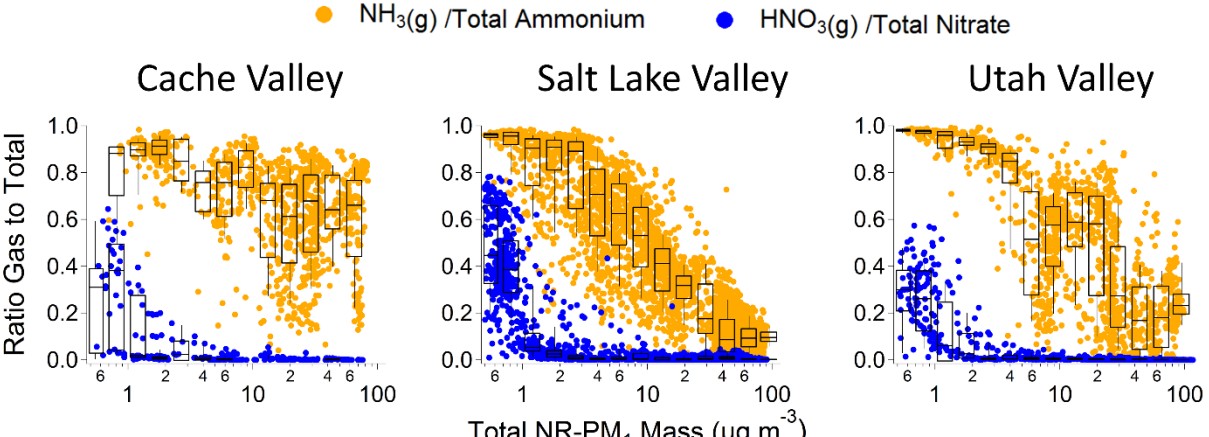

**Figure 9: Fraction of gas phase species to total mass (gas + aerosol) plotted as a function of total observed aerosol mass. In orange is the ratio between gas phase ammonia $NH_3$ and total ammonium $NH_3(g)+NH_4^+$. In blue is the ratio between gas phase nitric acid $HNO_3$ and total nitrate $HNO_3 + NO_3^-$. Overlaid onto the data point are box and whiskers plots for logarithmically spaced bins. The boundary of the boxes are 25[th] and 75[th] percentiles, the line in the box is the 50[th] percentile and the whiskers are 10[th] and 90[th] percentiles.**

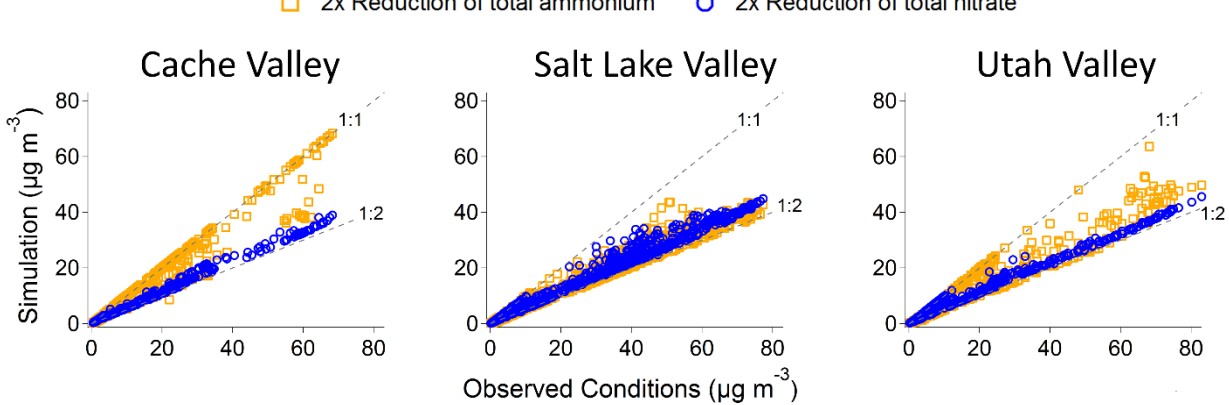

**Figure 10: Comparison of the simulated total inorganic aerosol mass (y-axis) to the observed total inorganic aerosol mass with separate reductions in precursor species using ISORROPIA (x-axis). Two different scenarios are shown: a twofold reduction of total ammonium (orange) and a twofold reduction in total nitrate (blue). The dashed grey lines in each plot represent the 1:1 line (i.e. no change in the simulated total inorganic mass in the aerosol phase) and the 1:2 line (i.e. a twofold reduction in the simulated total inorganic mass in the aerosol phase).**




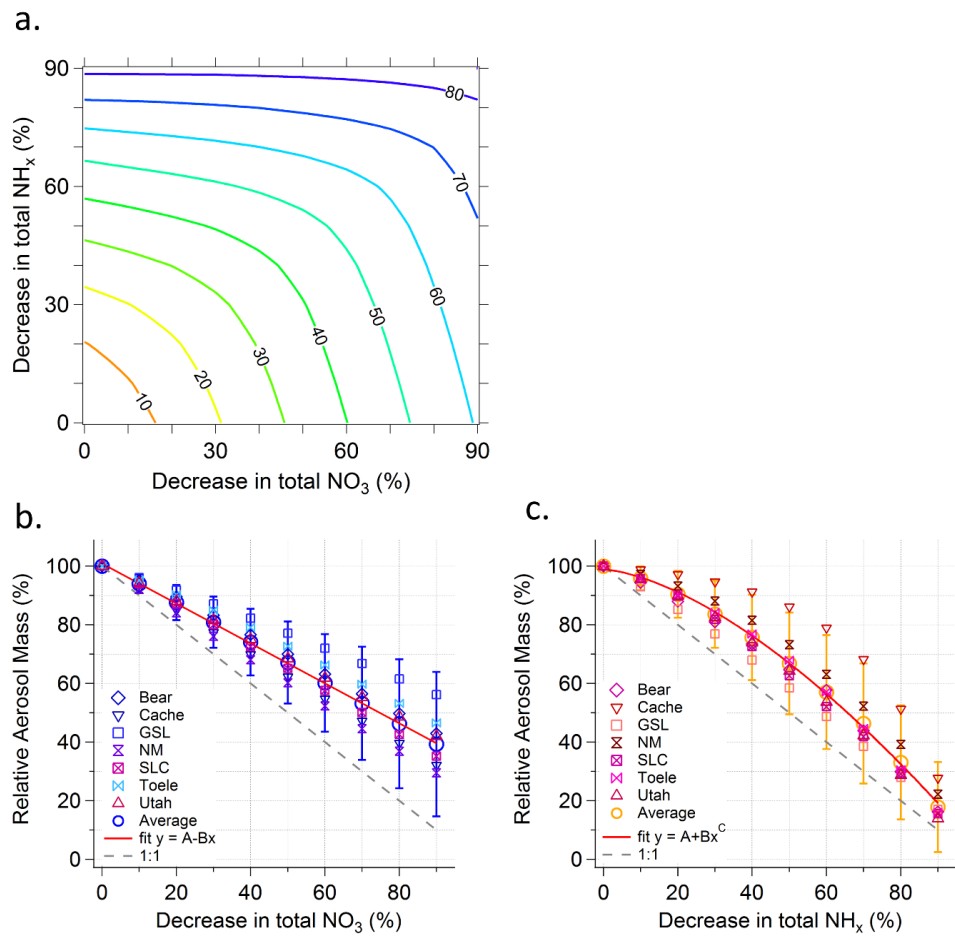

**Figure 11: a) Contour plot showing the simulated decrease in total inorganic aerosol mass (contour lines, numbers shown in %) as a function of both decrease in total nitrate (x-axis) and in total ammonium (y-axis). The data inputs used here are from the entire region. The decreases are relative to the observed conditions and expressed as percentages. Panels b) and c) show one-dimensional views of the surface plot shown in panel a). Panels b) and c) are projections along the x- and y-axes respectively. b) shows the simulated decrease in total inorganic NR-PM1 as a function of the decrease in total nitrate. In these simulations, the concentrations of total ammonium were kept constant, equal to the observed conditions. c) shows the simulated decrease in total inorganic NR-PM1 as a function of the decrease in total ammonium. In these simulations, the concentrations of total nitrate were kept constant; equal to the observed conditions. The open circles are averages over the entire region (same as panel a.). The error bars are the standard deviations associated to the averages. The other markers are data relative to particular regions: Bear Valley, Cache Valley, the Great Salt Lake, North Metro, Salt Lake Valley, Tooele Valley and Utah Valley. The dashed grey line is the 1:1 line and the red solid lines are a fit through the average points to guide the eye (linear fit in panel b.) and power law in panel c.)).**