# Peer review of "Airborne and ground-based observations of ammonium nitrate dominated aerosols in a shallow boundary layer during intense winter pollution episodes in northern Utah"

_Atmospheric Chemistry and Physics, 2018_

## Referee Comment (RC1) · Anonymous Referee #1 · 27 Sep 2018

The manuscript by Franchin et al. reports on airborne and ground-based gas and particle measurements during pollution episodes in Utah. The measurements focus mainly on the particle chemical composition (using the Aerodyne Aerosol Mass Spectrometer); the particles in the boundary layer mostly consist of ammonium nitrate (ca. 75% of PM1.0) during persistent cold-air periods with limited vertical mixing. Comparison between ammonia/ammonium and nitric acid/nitrate in the gas and particle phase reveals that the formation of ammonium nitrate was generally limited by the nitric acid concentration in the gas phase. Comparison between measurements and the calculations with the ISORROPIA model show generally good agreement. Simulations with the ISORROPIA model further indicate that a reduction in nitric acid can lead to an approximately proportional reduction in the aerosol mass loading, whereas the same is true for ammonia at 2 of the 3 measurement sites. The manuscript is well written and clearly structured. It shows very interesting data and indicates what knobs need to be turned in order to reduce the aerosol burden under the specified conditions. The comments I have are only minor and should be considered before the manuscript is published in ACP.

Page 2, line 3: delete "also"

Page 2, line 26: delete "and"

Page 4, line 11: replace "and" with "at"

Page 4, line 15: "laminar" instead of "linear" (see also page 5, line 16)

Page 5, line 29: Lee et al. (2017) is missing in the references

Page 5, line 30: "l min-1" instead of "liters per minute"

Page 6, line 5: please check the unit of the mass resolving power (usually expressed as Th/Th and not ppm)

Page 6, line 11: "measure"

Page 6, line 19/20: Do the authors mean by uncertainty the standard deviation of the signal at 1 Hz? Can you please also provide a value for the accuracy of the ammonia measurements?

Page 6, line 27: (i) If the offset at zero NO can be as high as 0.2 ppbv, is there a periodic zero measurement and correction performed? (ii) In this line the units used are pptv and ppbv; before the unit ppt was used, please use pptv consistently

Page 7, line 25: please check the use of the word "when"

Page 7, line 30: "It is . . ."

Page 7, line 30-32: since the AMS is not sensitive to particles < ∼70 nm, can the authors please also comment on the effects the exclusion of these small particles can have

Page 8, line 19: remove open bracket

Page 8, line 26: the green data points (Cache valley) show even higher values (up to 100 $\mu$g m-3)

Page 9, line 30: "C2H6"

Page 10, line 18: "Augsburg"?

Page 10, line 21: "compared with"

Page 11, line 11: delete "the"

Page 11, line 18: "emissions" instead of "concentrations"?

Page 12, line 5: "650 m AGL"

Page 12, line 24: "divided by"

Page 13, line 5: "than in Cache . . ."

Page 14, line 20: the 20% contour line seems to be rather yellow-greenish instead of orange

Page 14, line 26: I read the figure such that when a counter line intercepts with the maximum value on an axis, both ammonia and nitrate need to be reduced in order to decrease the aerosol loading further. This would be the case for > ∼60% regarding nitrate.

Figure 5: In most figures the unit $\mu$g m-3 is being used; it would be good not to switch between units (ppbv and $\mu$g m-3)

[Figure]

Figure 7: The agreement between UHSAS and AMS data is generally very good except for the bottom panel on the left. Is there any explanation why the concentrations differ in this profile?

SI (1st paragraph on page 1): Can the authors please specify what velocity they are referring to (particle velocity in the sampling line, velocity in the AMS flight chamber, . . .)?

---

## Referee Comment (RC2) · Anonymous Referee #2 · 7 Oct 2018

This paper deals with airborne and ground-based measurements of aerosol concentrations, chemical composition and gas phase precursors in three valleys in northern Utah (U.S.A.), obtained in winter 2017. It shows that increases in total aerosol mass "above $\sim 2~\mu gm-3$ were associated with increases in the ammonium nitrate mass fraction, clearly indicating that the highest aerosol mass loadings in the region were predominantly attributable to an increase in ammonium nitrate." The study shows a generally nitric acid ($HNO_3$) limited regime for the Cache and Utah Valleys, and a mixed regime ($HNO_3$ and $NH_3$) limited regime downwind of Salt Lake city. The inorganic species

were compared with the ISORROPIA thermodynamic model. Total inorganic aerosol mass concentrations were calculated for various decreases of total nitrate and total ammonium showing the combined total nitrate and total ammonium decreases were most efficient to reduce ammonium nitrate (so despite the prevailing HNO3 limitation).

The paper is a valuable contribution to ACP, because it gives a clear an sound estimate of the ammonium nitrate contribution to winter PM2.5 and of its limiting precursors.

The paper could be further improved if the following remarks were taken into account :

In the Introduction (page 3, line 5), the authors write "those studies suggested that nighttime nitrate formation within the residual layer was a major contributor to surface level PM2.5 concentrations." I understood that investigating this type of processes was one of the goals of the campaign. But little is said in the paper about the origin of ammonium nitrate precursors, within the study region, or outside. During winter time, kow qicly HNO3 would be formed from NOx emissions. May be the observational data set does not allow such a discussion, or it is foreseen for another paper. Authors should please state something about that. If the origin within the study region is most important, it would be interesting to show NOx and NH3 emission maps.

The authors state that other cations as Ca2+ or Mg2+ could bound a major part of nitrate. Although this is excluded of total nitrate as defined in the paper, it would be interesting to know what Ca2+ or Mg2+ levels could be typically expected.

How do meteorological conditions affect the isorropia calculated equilibria ? E4xtend the discussion would be interesting. Is aerosol expected to be liquid or solid ?

In Figure 7b , Salt lake city: The potential temperature profile decreases with altitude. This is not possible in this extent. Are potential temperature and temperature mismatched? In figure 8, another profile is shown.

Minor remarks :

Introduction: In Europe Petetin et al., 2016 performed a similar study as the one pre-

sented, although based on daily measurements and for an urban environment. This could be cited.

Page 5, line 4 : "It is important to note that refractory species such as sea and / or lake salt (mostly sodium chloride), road salt (mostly magnesium chloride), dust (mostly alkali salts and silicon oxides), and black carbon (from diesel exhaust or wood combustion) are not routinely measured with the AMS" The sentence is not clear. I think you mean that they have not been measured during the campaign.

Page 6, line 27: "but zero uncertainty can be as large as 0.2 ppbv" I wonder what are the reasons for this ? Could it be due to HNO3 sticking on inlet walls, and getting desorbed ?

Page 8, line 24 "During the first pollution episode, both the aircraft and ground-based aerosol mass concentrations were the highest in Cache Valley ($\sim$70 $\mu$g m-3) and the lowest in Utah Valley ($\sim$10 $\mu$g m-3). During the second pollution episode, the highest aerosol mass concentrations were observed in Utah Valley ($\sim$70 $\mu$g m-3) and varied for Salt Lake and Cache Valley over the course of the episode (40 – 90 $\mu$g m-3). During UWFPS, PM2.5 was on average around 70% of PM10 in the three valleys (Figure S3)." How to explain these differences between valley concentrations for different episodes be explained ?

Page 8, line 30: Please define Âń total aerosol mass Âż, probably PM1 non-refractive species measured by AMS.

Page 9, line 20: "The scatterplot gives an average slope of 1.8 g cm-3, which is comparable to the density of ammonium nitrate (1.72 g cm-3), within the combined measurement uncertainty." But it could be consistent, within the measurement uncertainty, with the density of other aerosol species too, isn't it ?

Page 9, line 23: "Those correspond to measurements in the plume of the U.S. Magnesium plant (Figure S1) where we observed a high organic (74.0$\pm$1.2%), high chloride

(6±0.6%) and low ammonium mass fraction (3±0.6%)." What are the typical emissions of such a factory ?

Page 13, line 27 "Reduction in total nitrate led to approximately a proportional decrease in total aerosol concentrations during polluted conditions and across all locations." Is total aerosol or total nitrate meant ?

Page 15, line 6 : "The PM2.5 5 mass loadings for the region varied from below the AMS detection limit of 0.38 to 72.4 $\mu$g m-3 (2nd and 98th percentiles)." This suggests that AMS measures PM2.5 or that PM2.5 was derived from AMS. Please clarify from which type of measurement PM2.5 loadings have been calculated here (Surface ? ).

Wording , typos :

Page 2, line 2: "lead" => "leads"

Page 5, line 28 : I would say Âń reactive nitrogen species Âż instead of "nitrogen oxides Âż. oxide

Page 8, line 19: Please correct Âń (, which is surrounded Âż

Page 9, line 30: Âń C2H6 Âż plus put indices.

Page 11, line 33 : Âń slightly Âż => "slight"

References : Petetin, H., Sciare, J., Bressi, M., Gros, V., Rosso, A., Sanchez, O., Sarda-Estève, R., Petit, J.-E., and Beekmann, M.: Assessing the ammonium nitrate formation regime in the Paris megacity and its representation in the CHIMERE model, Atmos. Chem. Phys., 16, 10419-10440, https://doi.org/10.5194/acp-16-10419-2016, 2016.

---

## Author Comment (AC1) · 4 Nov 2018

Answer to Referee # 1

We thank Anonymous Referee # 1 for their comments. Here below are the Authors' answers point-by-point (in blue) modifications to the text of the manuscript are reported in red.

Page 2, line 3: delete "also"
Deleted
Page 2, line 26: delete "and"
Deleted
Page 4, line 11: replace "and" with "at"
Replaced
Page 4, line 15: "laminar" instead of "linear" (see also page 5, line 16)
Replaced
Page 5, line 29: Lee et al. (2017) is missing in the references
The Referee is correct. We replaced "Lee et al. (2017)" with the correct reference "Lee et al (2018)"
Lee, B. H., Lopez-Hilfiker, F. D., Veres, P. R., McDuffie, E. E., Fibiger, D. L., Sparks, T. L., et al. (2018). Flight Deployment of a High-Resolution Time-of-Flight Chemical Ionization Mass Spectrometer: Observations of Reactive Halogen and Nitrogen Oxide Species. *Journal of Geophysical Research: Atmospheres*. https://doi.org/10.1029/2017JD028082
Page 5, line 30: "l min-1" instead of "liters per minute"
Changed
Page 6, line 5: please check the unit of the mass resolving power (usually expressed as Th/Th and not ppm)
The Referee is correct; the mas resolving power should be expressed in Th/Th. The mass accuracy of the instrument is usually better than 20 ppm. We changed the sentence from "The molecular formulae of the compounds listed above are readily identified given its mass resolving power (4500-5500 ppm (Junninen et al. 2010)..." to "The molecular formulae of the compounds listed above are readily identified given the instrument's mass resolving power (4500-5500 Th Th$^{-1}$ (Junninen et al. 2010)…"
Page 6, line 11: "measure"
Corrected
Page 6, line 19/20: Do the authors mean by uncertainty the standard deviation of the signal at 1 Hz? Can you please also provide a value for the accuracy of the ammonia measurements?
The uncertainty described here refers to the standard deviation of the 1 Hz signal. As it can be indeed misleading, we rephrased this sentence. Systematic errors were minimized by performing frequent backgrounds during flights. We added this information to the manuscript. "The uncertainty of the NH3 measurement during UWFPS was 150 ppt (1σ at 1 Hz sample frequency). To account for potential systematic errors, caused e.g., by changes in cabin temperature, zero measurements were performed regularly during flights "
Page 6, line 27: (i) If the offset at zero NO can be as high as 0.2 ppbv, is there a periodic zero measurement and correction performed? (ii) In this line the units used are pptv and ppbv; before the unit ppt was used, please use pptv consistently
A periodic zero was done for 30 seconds every 5-7 minutes. The text has been changed eliminating the inconsistency in units and adding the remrks from Refereee# 2. "The measurement accuracy was 5% for $O_3$, $NO_x$, and $NO_2$ and 12% for $NO_y$. Periodic zeros were measured for 30 s every 5-7 min. Measurements were less accurate during periods of rapid altitude change due to a minor pressure dependence in the background zeros in the NO channel that could not be fully corrected during post-processing."
Page 7, line 25: please check the use of the word "when"
Replaced "when" with "with"
Page 7, line 30: "It is . . ."
Corrected
Page 7, line x30-32: since the AMS is not sensitive to particles < _70 nm, can the authors please also comment on the effects the exclusion of these small particles can have
The majority of the mass is in the larger particles. We estimate from SMPS measurements carried out during UWFPS at the William Browning Building on the University of Utah campus site that particle with diameters

smaller than 70 nm contributed to less than 0.5% to the total mass. We changed the sentence from "It also important to note that the term "total nitrate" in this manuscript refers to gas phase plus $PM_1$ nitrate, but may exclude a non-negligible part of nitrate from the coarse mode." To "It also important to note that the term "total nitrate" in this manuscript refers to gas phase plus NR-$PM_1$ nitrate measured with an AMS. While we estimate that particles smaller than 70 nm (lower end of the transmission efficiency of the AMS) contributed to less than 0.5% to the total mass, with our definition we may exclude a non-negligible part of nitrate from the coarse mode."

Page 8, line 19: remove open bracket

Removed

Page 8, line 26: the green data points (Cache valley) show even higher values (up to 100 µg m-3)

Changed from "During the second pollution episode, the highest aerosol mass concentrations were observed in Utah Valley (~70 µg m-3) and varied for Salt Lake and Cache Valley over the course of the episode (40 – 90 µg m-3)" into "During the first pollution episode, both the aircraft and ground-based aerosol mass concentrations were the highest in Cache Valley (~70 µg m$^{-3}$ and ~90 µg m$^{-3}$ respectively) and the lowest in Utah Valley (~10 µg m$^{-3}$ and ~25 µg m$^{-3}$ respectively). During the second pollution episode, the highest mass concentrations observed at the ground sites in Cache Valley were up to 100 ug m$^{-3}$, in Utah Valley were ~70 ug m$^{-3}$, and in Salt Lake Valley were up to 60 ug m$^{-3}$. These variations among valleys in peak PM2.5 concentrations are characteristic and are due to variations in sources and meteorological processes (Baasandorj et al, 2018) (Figure S2)."

Page 9, line 30: "C2H6"

We changed the sentence from "… showed a contribution from organic fragments, probably $CH_2O$ or C2H6." To "… showed a contribution from organic fragments."

Page 10, line 18: "Augsburg"?

Corrected "Augsberg" to "Augsburg"

Page 10, line 21: "compared with"

Corrected

Page 11, line 11: delete "the"

Deleted

Page 11, line 18: "emissions" instead of "concentrations"?

Yes, we replaced "concentrations" with "emissions"

Page 12, line 5: "650 m AGL"

We added "m AGL" after "650"

Page 12, line 24: "divided by"

Corrected

Page 13, line 5: "than in Cache . . ."

Corrected

Page 14, line 20: the 20% contour line seems to be rather yellow-greenish instead of Orange

The referee is correct. We replaced "orange" with "yellow" in the text

Page 14, line 26: I read the figure such that when a counter line intercepts with the maximum value on an axis, both ammonia and nitrate need to be reduced in order to decrease the aerosol loading further. This would be the case for > _60% regarding nitrate.

We agree with the referee. We changed the sentence "However, both reagents must decrease in order to achieve a reduction of total aerosol mass larger than 40% relative to observed conditions." To "However, both reagents must decrease in order to achieve a reduction of total aerosol mass larger than 60% relative to observed conditions."

Figure 5: In most figures the unit µg m-3 is being used; it would be good not to switch between units (ppbv and µg m-3)

We changed the units to µg m$^{-3}$

Figure 7: The agreement between UHSAS and AMS data is generally very good except for the bottom panel on the left. Is there any explanation why the concentrations differ in this profile?

We improved the time alignment for UHSAS and AMS in a new version for Figure 7. The agreement between UHSAS and AMS in the bottom panel is within experimental uncertainties (Figure S6).

SI (1st paragraph on page 1): Can the authors please specify what velocity they are referring to (particle velocity in the sampling line, velocity in the AMS flight chamber, . . .)?

We modified the paragraph by adding the text in red: "Normal procedures were used to calibrate the AMS flow rate as a function of measured lens pressure and particle time-of-flight velocity (i.e. the velocity of the aerosol particles in vacuum, from the chopper to the vaporizer) as a function of particle size [Canagaratna et al., 2007]. For airborne measurements, we used a pressure-controlled inlet (PCI) that maintained a constant mass flow rate into the AMS [Bahreini et al., 2008]. Because particle time-of-flight velocity depends on … the PCI also provided a stable particle time-of-flight velocity calibration."

---

## Author Comment (AC2) · 4 Nov 2018

Answer to Referee # 2

We thank Anonymous Referee # 2 for their comments. Here below are the Authors' answers point by point )in blue).

In the Introduction (page 3, line 5), the authors write "those studies suggested that nighttime nitrate formation within the residual layer was a major contributor to surface level PM2.5 concentrations." I understood that investigating this type of processes was one of the goals of the campaign. But little is said in the paper about the origin of ammonium nitrate precursors, within the study region, or outside. During winter time, kow qicly HNO3 would be formed from NOx emissions. May be the observational data set does not allow such a discussion, or it is foreseen for another paper. Authors should please state something about that. If the origin within the study region is most important, it would be interesting to show NOx and NH3 emission maps.

The Referee is correct. Identifying the sources was indeed one of the objectives of the UWFPS measurement campaign. However, the scope of this manuscript was to 1) characterize the chemical composition of the aerosol particles and 2) investigate the effects of the decreases of total nitrate and total ammonium in the region. In other words "to achieve a better understanding of the processes that drive the conversion of precursor vapors into aerosol particles," as we wrote in the introduction (page 3 line 26).

Other manuscripts such as Moravek et al., (in preparation) will discuss the sources of NH3 in the region, showing the NH3 emission maps from the inventory and comparing them to the Twin Otter measurements. The paper will focus on $NH_3$ sources, but will also touch on NOx sources in the region. In addition, a second manuscript by McDuffie et al (in preparation) will investigate the contribution of nocturnal heterogeneous reactive nitrogen chemistry to particulate matter formation. A third paper by Womack et al (under review) explores NOx and VOC control as mitigation strategies for PM. Unfortunately, all these manuscripts are not published yet and therefore not possible to cite properly in the current paper.

The authors state that other cations as Ca2+ or Mg2+ could bound a major part of nitrate. Although this is excluded of total nitrate as defined in the paper, it would be interesting to know what Ca2+ or Mg2+ levels could be typically expected.

We have ion chromatography measurements of gas and aerosol phase at the University of Utah site that show presence of $Ca^{2+}$, $Na^+$ and $K^+$ in PM$_{2.5}$ ($Mg^{2+}$ was always close to the detection limit ~0.01 ug/m$^3$). The average concentrations were 0.14 $\mu g\ m^{-3}$ (max 1.2 $\mu g\ m^{-3}$) 0.09 $\mu g\ m^{-3}$ (max 3.6 $\mu g\ m^{-3}$) and 0.25 $\mu g\ m^{-3}$ (max 1.6 $\mu g\ m^{-3}$) respectively. The mass fractions of these species were always less than 10% of the total PM2.5 when the PM2.5 was > 20 ug m$^{-3}$). Ca2+ was always less than 3% and usually less than 1% for the high PM2.5 mass periods. The results of those measurements will be presented in detail in a separate manuscript by Hrdina et al. (in preparation).

How do meteorological conditions affect the isorropia calculated equilibria ? E4xtend the discussion would be interesting. Is aerosol expected to be liquid or solid ?

We agree with the Referee that it would be interesting to explore how the meteorological variables and aerosol phase affect the equilibria from the ISORROPIA model. However, we think that such a discussion is out of the scope of the paper. Cold temperatures and high relative humidity generally favor the

formation of ammonium nitrate in the aerosol phase in both the model and our observations. In the area studied during UWFPS the median RH and temperature were 0.66 and 271 K, respectively, and the phase of the aerosol is unknown at these conditions.

In Figure 7b , Salt lake city: The potential temperature profile decreases with altitude. This is not possible in this extent. Are potential temperature and temperature mismatched? In figure 8, another profile is shown.
The referee is correct the temperatures presented in Figures 7 are 8 air temperatures, not potential temperatures. We corrected the labels in both figures. In Figure 8 different profiles are shown because those are averages over polluted periods, not single missed approaches as in Figure 7.

Minor remarks :
Introduction: In Europe Petetin et al., 2016 performed a similar study as the one pre-sented, although based on daily measurements and for an urban environment. This could be cited.
We added the citation

Page 5, line 4 : "It is important to note that refractory species such as sea and / or lake salt (mostly sodium chloride), road salt (mostly magnesium chloride), dust (mostly alkali salts and silicon oxides), and black carbon (from diesel exhaust or wood combustion) are not routinely measured with the AMS" The sentence is not clear. I think you mean that they have not been measured during the campaign.
We changed the sentence into "It is important to note that during the campaign refractory species such as sea and / or lake salt (mostly sodium chloride), road salt (mostly magnesium chloride), dust (mostly alkali salts and silicon oxides), and black carbon (from diesel exhaust or wood combustion) were not measured."

Page 6, line 27: "but zero uncertainty can be as large as 0.2 ppbv" I wonder what are the reasons for this ? Could it be due to HNO3 sticking on inlet walls, and getting desorbed?
The reasons to the uncertainty being so large at times is related to a minor pressure-dependence in the background zeros in the NO channel. We added that information in the text. Inlet effects, such as HNO3 sticking on inlet walls, would not be relevant for the NO, NO2, O3 channels, as those channels are "blind" to HNO3.

Page 8, line 24 "During the first pollution episode, both the aircraft and ground-based aerosol mass concentrations were the highest in Cache Valley (_70 µg m-3) and the lowest in Utah Valley (_10 µg m-3). During the second pollution episode, the highest aerosol mass concentrations were observed in Utah Valley (_70 µg m-3) and varied for Salt Lake and Cache Valley over the course of the episode (40 – 90 µg m-3). During UWFPS, PM2.5 was on average around 70% of PM10 in the three valleys (Figure S3)."
How to explain these differences between valley concentrations for different episodes be explained ?
We added at page 9 line 5: ""These variations among valleys in peak PM2.5 concentrations are characteristic and are due to variations in sources and meteorological processes (Baasandorj et al, 2018) (Figure S2)."
We further modified the prior sentences, taking surface based measurements during non-flight times into account, to: "During the first pollution episode, both the aircraft and ground-based aerosol mass concentrations were the highest in Cache Valley (~70 µg m-3 and ~90 µg m-3 respectively) and the lowest in Utah Valley (~10 µg m-3 and ~25 µg m-3 respectively). During the second pollution episode, the highest mass concentrations observed at the ground sites in Cache Valley were up to 100 ug m-3, in Utah Valley were ~70 ug m-3, and in Salt Lake Valley were up to 60 ug m-3."
Page 8, line 30: Please define ´n total aerosol mass ˙z, probably PM1 non-refractory species measured by AMS.
We modified the sentence to "For the purposes of this analysis, we define clean conditions, when the total aerosol mass (NR-PM$_1$) is ≤ 2 µg m$^{-3}$ and …"

Page 9, line 20: "The scatterplot gives an average slope of 1.8 g cm-3, which is comparable to the density of ammonium nitrate (1.72 g cm-3), within the combined measurement uncertainty." But it could be consistent, within the measurement uncertainty, with the density of other aerosol species too, isn't it ?

That is correct. However, ammonium nitrate is mentioned here because it is the largest component of the aerosol as the AMS measurements show.

Page 9, line 23: "Those correspond to measurements in the plume of the U.S. Magnesium plant (Figure S1) where we observed a high organic (74.0±1.2%), high chloride (6±0.6%) and low ammonium mass fraction (3±0.6%)." What are the typical emissions of such a factory ?

The U.S. magnesium plant it is the largest producer of primary magnesium in North America. It has a natural gas power plant on site and a processing plant. According to our measurements, it emits a large amount of aerosols, chloride and other halogens in the gas phase.

Page 13, line 27 "Reduction in total nitrate led to approximately a proportional decrease in total aerosol concentrations during polluted conditions and across all locations." Is total aerosol or total nitrate meant ?

As written, we meant total inorganic aerosol. We changed the sentence to be more precise into "Reduction in total nitrate led to approximately a proportional decrease in total inorganic aerosol concentrations during polluted conditions and across all locations"

Page 15, line 6 : "The PM2.5 mass loadings for the region varied from below the AMS detection limit of 0.38 to 72.4 µg m-3 (2nd and 98th percentiles)." This suggests that AMS measures PM2.5 or that PM2.5 was derived from AMS. Please clarify from which type of measurement PM2.5 loadings have been calculated here (Surface ? ).

The referee is correct. It should be NR-PM$_1$. We corrected the sentence.

Wording , typos :
Page 2, line 2: "lead" => "leads"
Corrected

Page 5, line 28 : I would say ´n reactive nitrogen species Â˙z instead of "nitrogen oxides Â˙z. oxide

We decided to leave the sentence unchanged, as by using "reactive nitrogen species" the sentence would result less specific.

Page 8, line 19: Please correct ´n (, which is surrounded Â˙z
Corrected

Page 9, line 30: ´n C2H6 Â˙z plus put indices.

We changed the sentence from "… showed a contribution from organic fragments, probably CH$_2$O or C2H6." To "… showed a contribution from organic fragments."

Page 11, line 33 : ´n slightly Â˙z => "slight"
Corrected

References
Baasandorj, M., Brown, S. S., Hoch, S., Crosman, E., Long R., Silva, P. Mitchell L., Hammond I., Martin, R., Bares R., Lin, J., Sohl J.,Page, J., McKeen, S., Pennell, C., Franchin, A., Middlebrook, A., Petersen, R., Hallar, G., Fibiger, D., Womack, C., McDuffie, E., Moravek, A., Murphy, J., Hrdina, A., Thornton, J., Goldberger, L., Lee, B., Riedel, T., Whitehill, A., Kelly, K., Hansen, J., Eatough, D., 2017 Utah Winter Fine Particulate Study Final Report, 2017.https://www.esrl.noaa.gov/csd/groups/csd7/measurements/2017uwfps/finalreport.pdf (Last accessed June 2018)